# Spatial Conformal Inference through Localized Quantile Regression

## Abstract

Reliable uncertainty quantification at unobserved spatial locations, particularly for complex and heterogeneous datasets, is a key challenge in spatial statistics. Traditional methods like Kriging rely on strong distributional assumptions such as normality, which often fail in large-scale datasets, leading to unreliable intervals. Conformal prediction offers distribution-free coverage, yet most implementations rely on i.i.d. data and ignore spatial dependence. We propose Localized Spatial Conformal Prediction (LSCP), a model-agnostic framework that couples local quantile regression with conformal calibration to produce spatially adaptive prediction intervals. LSCP conditions on neighborhoods in space to capture local heterogeneity and relaxes i.i.d. requirements: it retains finite-sample marginal coverage under exchangeability and, under stationarity and spatial mixing, attains asymptotic conditional coverage. Across synthetic and real datasets, LSCP consistently achieves near-nominal coverage with tighter and more stable intervals than other existing CP methods.

## 1 Introduction

Quantifying uncertainty at unobserved spatial locations has been a longstanding challenge in spatial statistics, particularly in practical applications such as weather forecasting (Siddique et al., 2022) and mobile signal coverage estimation (Jiang et al., 2024). Traditional methods like Kriging rely on strong parametric assumptions, including normality and stationarity, to model spatial relationships and quantify uncertainty (Cressie, 2015). However, the failure of these assumptions in the complex spatial datasets (Heaton et al., 2019) results in unreliable uncertainty quantification. The issue is especially pronounced when constructing prediction intervals, as deviations from stationarity or Gaussianity can critically undermine their validity (Fuglstad et al., 2015).

While many methods have been developed to handle heterogeneity (Gelfand et al., 2005; Duan et al., 2007), these approaches are often computationally expensive and cannot scale effectively for massive datasets. Furthermore, fully modeling the underlying process is not always necessary, particularly when the goal is to produce reliable prediction intervals. Recently, machine learning approaches have offered alternative strategies for spatial prediction (Hengl et al., 2018; Chen et al., 2020), though they tend to focus on point predictions and often lack rigorous uncertainty quantification.

Conformal prediction, introduced by (Vovk et al., 2005), provides a powerful, distribution-free approach to uncertainty quantification. Its ability to generate valid prediction sets without assumption on the underlying data distribution and the prediction model has gained widespread popularity in both machine learning and statistics (Lei & Wasserman, 2014; Angelopoulos et al., 2023). By leveraging only the exchangeability of data, conformal prediction ensures valid coverage at any significance level, making it highly attractive for scenarios where a black-box model is used, or traditional parametric assumptions may fail.

However, in many real-world datasets—such as time series data—the assumption of exchangeability does not hold. To address this, recent work has extended conformal prediction to handle non-exchangeable data. For instance, (Tibshirani et al., 2019) introduced weighted quantiles to maintain valid coverage in the presence of distributional shifts between training and test sets by leveraging the likelihood ratio between distributions. More recently, (Barber et al., 2023) tackled the challenge of distribution shifts by bounding the coverage gap using the total variation distance, although the issue

of optimizing these weights remains open. For time series data, further improvements have been made in tightening prediction intervals and building theoretical guarantees, as demonstrated by (Xu & Xie, 2023; Xu et al., 2024).

In this paper, we extend conformal prediction to spatial data, where the assumption of exchangeability rarely holds. While time series data can be viewed as a special case of spatial data defined in a one-dimensional time domain, spatial data is inherently multidimensional and poses unique challenges. For example, while time indices are typically discrete and naturally ordered, spatial locations are continuous and lack intrinsic ordering. Despite the prevalence of spatial data in real-world applications, there has been limited work on conformal prediction methods tailored to this context. To address this gap, we propose Localized Spatial Conformal Prediction (LSCP), a novel conformal prediction method that employs localized quantile regression for constructing prediction intervals. Our method and theoretical framework can also be extended to spatio-temporal settings, broadening its applicability. Here is a revised version of the summary:

- *Localized Spatial Conformal Prediction (LSCP)*: We introduce LSCP, a conformal prediction algorithm specifically designed for spatial data, that learns data-adaptive localization via quantile regression on neighborhood residuals, yielding intervals that adapt to complex spatial dependence.

- *Theoretical guarantees:* We establish a finite-sample bound for the coverage gap and provide asymptotic convergence guarantees for LSCP, without requiring the exchangeability of the data. When exchangeability holds, LSCP also enjoys the standard distribution-free marginal coverage guarantee of traditional CP methods.

- *Numerical Evaluation:* We extensively evaluate LSCP against baseline CP methods on both synthetic and real-world datasets. The results highlight the ability of LSCP to achieve tighter prediction intervals with valid coverage and more uniform calibration across the spatial domain.

## 1.1 LITERATURE

*Conformal prediction beyond exchangeability.* Traditional conformal prediction relies on the assumption of exchangeability, which is often violated in real-world scenarios such as time series forecasting. To address this, recent research has extended the framework to non-exchangeable settings, primarily through weighted conformal prediction. For example, Tibshirani et al. (2019) addressed covariate shift by weighting samples by the likelihood ratio, a concept generalized by Barber et al. (2023) to bound coverage gaps using total variation distance. In the specific context of time series, methods often assign higher weights to recent data (Xu et al., 2024) or adaptively adjust the significance level $\alpha$ to correct for distribution drift (Gibbs & Candes, 2021; Angelopoulos et al., 2024). However, finding a universally optimal weighting strategy remains an open challenge (Barber et al., 2023). Furthermore, in the absence of exchangeability, finite-sample guarantees are typically replaced by asymptotic validity, achieved under specific mixing or stability assumptions.

*Conformal prediction for spatial and localized data.* The spatial setting represents a broader and more complex domain compared to time series data, yet research on conformal prediction for spatial contexts remains limited. A recent study by (Mao et al., 2024) introduced a spatial conformal prediction method under the infill sampling framework, employing kernel functions to weight conformity scores based on spatial proximity. This approach aligns with a broader class of localized conformal methods designed to approximate conditional coverage. For instance, (Guan, 2023) and (Han et al., 2022) proposed general frameworks for localized conformal prediction, utilizing kernel functions to adapt prediction intervals to the local feature space. More recently, theoretical advances have formalized these conditional goals: (Gibbs et al., 2025) established fundamental limits and methods for achieving coverage over specific covariate shifts, while (Colombo, 2024) and (Plassier et al., 2024) leveraged generative models and conditional density estimators to construct flexible, locally adaptive prediction sets with approximate conditional validity. While not all explicitly designed for spatial data, these methods highlight the critical role of local weighting and conditional estimation in handling complex, non-exchangeable data structures.

## 2 PROBLEM SETUP

In this paper, we consider a spatial setting with observations $\{Z(s_i)\}_{i=1}^n$, where $Z(s) = (X(s), Y(s))$ represents a random field observed at spatial locations $s_i$. Here, $Y(s) \in \mathbb{R}$ is the response variable, and $X(s) \in \mathbb{R}^p$ is the associated feature vector, potentially including spatial location $s$. In the paper, we follow the commonly-used stochastic design in spatial setting, which means that the spatial locations $s$ are i.i.d. samples from an unknown distribution $g(s)$.

In conformal prediction, the objective is to construct a prediction region $\hat{C}_n(X(s_{n+1}))$ for an unobserved response $Y(s_{n+1})$ given a known feature vector $X(s_{n+1})$. For a user-specified confidence level $\alpha$, we aim to ensure that the probability of $Y(s_{n+1})$ falling within the prediction region exceeds $1 - \alpha$. This notion of coverage can be interpreted in two ways: marginal coverage and conditional coverage. *Marginal coverage* is defined as

$$\mathbb{P}(Y(s_{n+1}) \in \hat{C}_n(X(s_{n+1}))) \geq 1 - \alpha,$$

whereas *conditional coverage* requires that

$$\mathbb{P}(Y(s_{n+1}) \in \hat{C}_n(X(s_{n+1})) \mid X(s_{n+1})) \geq 1 - \alpha.$$

Conditional coverage is stricter than marginal coverage, requiring validity for all $X(s)$. However, as shown by (Foygel Barber et al., 2021), achieving universal conditional coverage is impossible without additional assumptions. In standard conformal settings, where data is i.i.d. or exchangeable, only marginal coverage is typically guaranteed.

## 3 METHOD

We first provide the necessary background on the conventional split conformal prediction framework, which yields prediction intervals of uniform width. Building upon this, we introduce our proposed Localized Spatial Conformal Prediction (LSCP) method, designed to generate more adaptive intervals by modeling spatial correlation. We situate our contribution by comparing with several related techniques in recent literature: Global Spatial Conformal Prediction (GSCP), Smoothed Localized Spatial Conformal Prediction (SLSCP) from (Mao et al., 2024), and Localized Conformal Prediction (LCP) from (Guan, 2023). A detailed discussion is in Appendix B and Table 1 summarizes the differences.

### 3.1 BACKGROUND: CONFORMAL PREDICTION

Conformal prediction constructs a prediction interval for $Y_{n+1}$ given a prediction model $\hat{f}$, feature $X_{n+1}$ and past observations $\{(X_i, Y_i)\}_{i=1}^n$. The prediction model $\hat{f}$ can be any user-specified model. The data is split into a training set to fit $\hat{f}$ and a calibration set to compute non-conformity scores, typically $\hat{\varepsilon}_i = |Y_i - \hat{f}(X_i)|$. Using the empirical quantile $\widehat{Q}_n(1 - \alpha)$ of these scores, the prediction interval is constructed as:

$$\hat{C}_n(X_{n+1}) = [\hat{f}(X_{n+1}) - \widehat{Q}_n(1 - \alpha), \ \hat{f}(X_{n+1}) + \widehat{Q}_n(1 - \alpha)].$$

Conformal prediction ensures valid marginal coverage of $1 - \alpha$ under exchangeability and is flexible, requiring no assumptions on the distribution of $Y$ or the form of $\hat{f}$.

In contrast, traditional geospatial methods like Gaussian Processes (GPs) assume a Gaussian prior with explicit covariance structure, providing mean predictions and uncertainty estimates. While GPs are powerful, their reliance on Gaussian assumptions can lead to poor coverage when data deviates from this distribution. GPs are also computationally expensive, limiting scalability to large datasets. Conformal prediction, by contrast, is computationally efficient, model-agnostic, and provides finite-sample coverage guarantees, making it more practical in many scenarios.

### 3.2 PROPOSED METHOD: LOCAL SPATIAL CONFORMAL PREDICTION (LSCP)

In spatial settings, data often exhibit significant dependence across locations, and taking the spatial dependence into account can greatly improve the accuracy of prediction intervals. To account for this,

$$(\varepsilon_1, \cdots, \varepsilon_n) \stackrel{d}{=} (\varepsilon_{\sigma(1)}, \cdots, \varepsilon_{\sigma(n)})$$

Exchangeability

$$(\varepsilon(s_1), \cdots, \varepsilon(s_n)) \stackrel{d}{=} (\varepsilon(s_1 + \delta), \cdots, \varepsilon(s_n + \delta)), \forall \delta$$

Spatial Stationarity

Figure 1: Difference between exchangeability and spatial stationarity. Exchangeability means permutation-invariant while stationarity represents translation-invariant.

it is advantageous to base predictions on nearby data points, as spatially proximate observations are likely to share similar distributions and therefore provide more reliable information. The recent study by (Barber et al., 2023) highlights the importance of weighting calibration data differently, depending on their relevance to the target prediction point.

Assume the calibration set consists of observations $(X(s_1), Y(s_1)), \ldots, (X(s_n), Y(s_n))$. For a new observation $(X(s_{n+1}), Y(s_{n+1}))$, the aim is to construct a prediction interval by selecting a neighborhood of data from the calibration data. Here we use $N(s_{n+1})$ to represent the neighborhood of $s_{n+1}$, which can be determined via various criteria, and a common approach is to include all nearby points located within a specified distance threshold. In the paper, we use $k$-nearest neighbors for simplicity.

Given a pretrained prediction model $\hat{f}$, the non-conformity scores are defined as $\hat{\varepsilon}(s_i) = Y(s_i) - \hat{f}(X(s_i))$. Let $\mathcal{E}_{n+1} = \{\hat{\varepsilon}(s_i)\}_{i \in N(s_{n+1})}$ be the non-conformity scores of the neighbors of $s_{n+1}$, we then define the conditional cumulative distribution function (CDF), denoted by

$$F(e|\mathcal{E}_{n+1}) = \mathbb{P}(\hat{\varepsilon}(s_{n+1}) \leq e|\mathcal{E}_{n+1}).$$

The conditional quantile $Q_n(p)$ is defined as

$$Q_n(p) = \inf\{e \in \mathbb{R} : F(e|\mathcal{E}_{n+1}) \geq p\}. \tag{1}$$

Let $\widehat{Q}_n(p)$ be an estimator of the true quantile $Q_n(p)$ in Equation 1, the prediction interval of LSCP is defined as:

$$\hat{C}_n(X(s_{n+1})) = [\hat{f}(X(s_{n+1})) + \widehat{Q}_n(\beta^*), \ \hat{f}(X(s_{n+1})) + \widehat{Q}_n(1 - \alpha + \beta^*)],$$

where $\beta^* = \operatorname{argmin}_{\beta \in [0,\alpha]}(\widehat{Q}_n(1 - \alpha + \beta) - \widehat{Q}_n(\beta))$. Here $\beta$ is optimized to find the tightest interval. In particular, if we choose $\widehat{Q}_n(p)$ to be the empirical quantile and $\beta^* = \alpha/2$, then SPCI reduces to a localized version of GSCP.

In order to leverage the dependency in residuals and produce intervals as narrow as possible, we apply a quantile regression estimator $\widehat{Q}_n$ to predict the conditional quantile of the residual at $s_{n+1}$ based on its neighboring residuals $\{\hat{\varepsilon}(s_i)\}_{i \in N(s_{n+1})}$. For computational efficiency, we use Quantile Random Forests (QRF) from (Meinshausen & Ridgeway, 2006), although other quantile regression techniques could also be applied. The QRF learns the conditional quantile by treating the residual at location $s_{n+1}$, denoted $\tilde{Y}(s_{n+1}) = \hat{\varepsilon}(s_{n+1})$, as a function of the vector of its neighboring residuals, $\tilde{X}(s_{n+1}) = (\hat{\varepsilon}(s_i))_{i \in N(s_{n+1})}$. Here we order the residuals by increasing spatial distance $\|s_i - s_{n+1}\|$ to preserve the local dependence structure. This consistency is critical for leveraging the efficiency of QRF. The estimator yielded by the QRF is a weighted empirical quantile:

$$\widehat{Q}_n(p) = \inf\{e \in \mathbb{R} : \sum_{i=1}^n \omega_i \mathbf{1}\{\tilde{Y}(s_i) \leq e\} \leq p\},$$

where the weights $\omega_i$ are learned through quantile regression. The derivation is detailed in Appendix A.1. One key difference between LSCP and other spatial conformal prediction method lies in that the weight $\omega_i$ is data-adaptive instead of user-specified and Table 1 summarizes the differences.

## 4 THEORETICAL RESULTS

### 4.1 NOTATIONS

Suppose the data is denoted by $\{Z(s_i)\}_{i=1}^n$, where $Z(s) = (X(s), Y(s))$, $s \in \mathbb{R}^d$ denotes the spatial location, $X(s) \in \mathbb{R}^d$ is the feature vector and $Y(s) \in \mathbb{R}$ represents the univariate response. We

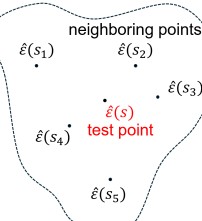 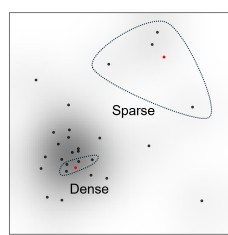

Figure 2: Illustration of neighborhoods in Localized Spatial Conformal Prediction (LSCP). Left: $k$-nearest neighbors are used as features to capture spatial dependence. Right: neighborhoods adapt to both dense and sparse regions. Red points denote test locations, dashed circles indicate the 5-nearest neighbors, and darker shading corresponds to lower uncertainty.

Table 1: Comparison of assumptions and algorithms across localized conformal prediction methods.

|  | **LSCP** (Ours) | **SLSCP** (Mao et al., 2024) | **LCP** (Guan, 2023) |
|---|---|---|---|
| Algorithm | Weighted quantile via learned quantile regression | Weighted quantile via fixed kernel function | Weighted quantile via fixed kernel function |
| Weights | Data-adaptive, learned by quantile regression on residual features | User-specified kernel based on spatial distance | User-specified kernel based on feature similarity |
| Dependence captured | Complex dependence learned directly from local residual patterns | Spatial dependence limited to pairwise proximity via a given kernel | Feature dependence limited to pairwise similarity via a given kernel |
| Distributional assumptions | Stationary, spatially mixing noise | Infill sampling with locally i.i.d. noise | Globally i.i.d. data |
| Data model | Additive noise model | Continuous mapping from separate spatial and noise processes | No explicit structural assumption, but requires i.i.d. data |

assume that $Y(s)$ is generated from a true model with unknown additive noise:

$$Y(s) = f(X(s)) + \varepsilon(s),$$

where $f$ is an unknown function and $\varepsilon(s)$ represents the noise process, whose marginal distribution is not necessarily Gaussian. Given a pre-trained prediction model $\hat{f}$, we can compute the non-conformity scores

$$\hat{\varepsilon}(s) = Y(s) - \hat{f}(X(s)).$$

The estimated CDF $\widehat{F}(y)$ is defined as

$$\widehat{F}_{n+1}(y) = \sum_{i=1}^{n} w_i \mathbf{1}(\hat{\varepsilon}(s_i) \leq y),$$

where $\omega_i$ is the weight assigned to data point $s_i$ that satisfies $\sum_{i=1}^{n} \omega_i = 1$. Besides, we define the CDF and the weighted empirical CDF for true noise respectively as $F_\varepsilon(y)$ and $\widetilde{F}_{n+1}(y)$, where

$$\widetilde{F}_{n+1}(y) = \sum_{i=1}^{n} w_i \mathbf{1}(\varepsilon(s_i) \leq y).$$

## 4.2 PRELIMINARY

*Spatial mixing.* To derive theoretical guarantees, we must quantify the extent to which the random field $Z(\cdot)$ exhibits dependence, which means how quickly variables become statistically independent as the distance between them increases.

First, we define a measure of dependence between two specific spatial regions. Let $\mathcal{F}_Z(T) = \sigma \langle Z(s) : s \in T \rangle$ denote the $\sigma$-algebra generated by the random field on a subset $T \subset \mathbb{R}^d$. For any two disjoint regions $T_1, T_2 \subset \mathbb{R}^d$, the strong mixing coefficient between them is defined as the maximum difference between the joint probability and the product of marginal probabilities:

$$\tilde{\alpha}(T_1, T_2) = \sup \left\{ |\mathbb{P}(A \cap B) - \mathbb{P}(A)\mathbb{P}(B)| : A \in \mathcal{F}_Z(T_1), B \in \mathcal{F}_Z(T_2) \right\}.$$

---

**Algorithm 1** Localized Spatial Conformal Prediction

---

**Require:** Dataset $\{(x(s_i), y(s_i))\}_{i=1}^n$, prediction algorithm $\mathcal{A}$, significance level $\alpha$

**Ensure:** Prediction intervals $\{\widehat{C}_n(x(s_{n+1}))\}$

1: Split the dataset into training data and calibration data.
2: Train the prediction model $\hat{f}$ with the training data using the prediction algorithm $\mathcal{A}$.
3: Select the neighborhood of $s_{n+1}$ in the calibration data, denoted as $N(s_{n+1})$.
4: Compute the non-conformity scores $\hat{\varepsilon}(s_i) = Y(s_i) - \hat{f}(X(s_i))$ for all data in the calibration set.
5: Set $\tilde{Y}(s_i) = \hat{\varepsilon}(s_i)$ and $\tilde{X}(s_i) = (\hat{\varepsilon}(s_{j_1}), \ldots, \hat{\varepsilon}(s_{j_{|N(s_i)|}}))$, where $s_j \in N(s_i)$.
6: Fit quantile regression $\widehat{Q}_n$ with all pairs $(\tilde{X}(s_i), \tilde{Y}(s_i))$ in the calibration data.
7: Obtain the prediction interval $\hat{C}_n(X(s_{n+1}))$.

---

Intuitively, $\tilde{\alpha}(T_1, T_2)$ measures the strongest correlation between any event occurring in region $T_1$ and any event in region $T_2$.

Next, we define the mixing coefficient for the entire process. Unlike time series ($d = 1$), where past and future are clearly defined half-lines, spatial sets can take complex shapes. To handle this, we adopt the framework established by (Lahiri, 2003), which defines the mixing coefficient $\alpha(a; b)$ based on the worst-case dependence between sets separated by a distance $a$ with volume bounded by $b$.

Let $d(T_1, T_2) = \inf\{|x - s| : x \in T_1, s \in T_2\}$ be the minimum distance between two sets. We restrict our attention to $\mathcal{R}_k(b)$, the collection of all sets formed by the union of at most $k$ disjoint cubes in $\mathbb{R}^d$ with a total volume bounded by $b$:

$$\mathcal{R}_k(b) \equiv \left\{ \cup_{i=1}^k D_i : \sum_{i=1}^k |D_i| \leq b \right\}.$$

The strong-mixing coefficient $\alpha(a; b)$ for the random field is then defined as:

$$\alpha(a; b) = \sup\left\{\tilde{\alpha}(T_1, T_2) : d(T_1, T_2) \geq a, \, T_1, T_2 \in \mathcal{R}_3(b)\right\}.$$

Here, the parameter $a$ controls the *separation distance*, and $b$ controls the *volume* of the regions.

Finally, to ensure valid coverage, we assume this coefficient satisfies a standard decay condition: it must decay polynomially with distance while growing at most polynomially with volume. Specifically, we posit the existence of a non-increasing function $\alpha_1(\cdot)$ with $\lim_{a \to \infty} \alpha_1(a) = 0$ and a non-decreasing function $g(\cdot)$ such that:

$$\alpha(a; b) \leq \alpha_1(a)g(b), \quad \text{for all } a > 0, b > 0.$$

### 4.3 ASSUMPTIONS

With the notation above, we now state the assumptions that underlie our theoretical analysis. When the data are exchangeable, valid marginal coverage follows directly if using the empirical quantile as the quantile estimate. Our focus, however, is the non-exchangeable setting, common under localization and spatial dependence, where we replace exchangeability with stationarity and spatial mixing to establish coverage guarantees.

**Assumption 4.1** (Weight decay). There exists constant $\gamma > 0$ so that the normalized weights satisfy

$$\omega_i = o(n^{-\frac{1+\gamma}{2}}), \tag{2}$$

for all $i$; meaning that $M_n = \max_{1 \leq i \leq n} \omega_i = o(n^{-\frac{1+\gamma}{2}})$.

The requirement assumes that the normalized weights decay at a rate faster than $n^{-\frac{1}{2}}$. We emphasize that this assumption does not require neighborhood size $k$ to grow since the QRF is trained on the whole calibration dataset. As we can see, $\omega_i = \frac{1}{n}$ is a special case that satisfies this condition. The condition is also weaker than the requirement of $\omega_i = O(\frac{1}{n})$ in a related study (Xu et al., 2024). Besides, the assumption can also be inferred from that of SLSCP (Mao et al., 2024) where a GBF kernel with infinitely close data leads to uniform weights. Besides, the assumption automatically holds for QRF when setting the minimal samples per leaf to be $\lceil cn^\eta \rceil$ for constant $c > 0, \eta > \frac{1+\gamma}{2}$.

**Assumption 4.2** (Estimation quality). There exists a sequence $\{\delta_n\}_{n\geq 1}$ such that

$$\sum_{i=1}^{n} \|\hat{\varepsilon}(s_i) - \varepsilon(s_i)\|^2 \leq \frac{\delta_n^2}{M_n}, \tag{3}$$

$$\|\hat{\varepsilon}(s_{n+1}) - \varepsilon(s_{n+1})\| \leq \delta_n.$$

The assumption requires the average prediction error to be bounded by $\delta_n^2$, a weaker condition than in (Xu & Xie, 2021). Notably, our coverage gap result does not require $\delta_n$ to converge to zero, although this occurs in many settings. For example, research on neural network prediction error, such as (Barron, 1994), shows that under certain regularization conditions on $f$, $\delta_n = O\left(1/\sqrt{n}\right)$.

**Assumption 4.3** (Stationary and spatial mixing). The random field $\varepsilon(s)$ is stationary and strongly mixing with coefficient $\alpha$, and the strong mixing coefficient can be bounded by $\alpha(a,b) \leq \alpha_1(a)g(b)$, where $\alpha_1$ is a nonincreasing function with $\lim_{a\to\infty} \alpha_1(a) = 0$. We assume $E_{d\sim g_n}\alpha_1(d)^2 \leq \frac{M}{n^2}$ where $g_n(d)$ is distribution of the distance between two sample points $s_i$ and $s_j$ ($1 \leq i,j \leq n$).

The assumption requires the true noise $\varepsilon(s)$ to be spatially stationary and strongly mixing, a weaker condition than the i.i.d. assumption in (Mao et al., 2024). This accommodates complex dependencies and non-stationarity in the random field $\{X(s), Y(s)\}$. We assume a standard "decay with distance": the mixing coefficient $\alpha_1(d)$ is nonincreasing in the separation $d$ and vanishes as $d \to \infty$. This is the natural multi-dimensional extension of the strong-mixing condition commonly used in time series analysis.

**Assumption 4.4.** (Lipschitz continuity) The true CDF $F_\varepsilon(y)$ of noise $\varepsilon$ is assumed to be Lipschitz continuous with parameter $L_{n+1}$.

## 4.4 FINITE-SAMPLE MARGINAL COVERAGE

When we use empirical quantile as the quantile estimator for Equation 1, LSCP reduces to GSCP method. When spatial locations $s_i$ are i.i.d., then the spatial data $(X(s_i), Y(s_i))$ are exchangeable, and the finite-sample marginal coverage can be proved in a standard way.

**Lemma 4.5** (Finite-sample marginal coverage (Mao et al., 2024)). *When $s_1, \cdots, s_{n+1}$ are i.i.d., then $(X(s_1), Y(s_1)), \cdots, (X(s_{n+1}), Y(s_{n+1}))$ is an exchangeable sequence and*

$$\mathbb{P}(Y(s_{n+1}) \in \widehat{C}_n(X(s_{n+1}))) \geq 1 - \alpha. \tag{4}$$

## 4.5 CONDITIONAL COVERAGE

When the quantile is estimated by a weighted empirical distribution (e.g., via QRF), the symmetry is lost, so exchangeability no longer implies finite-sample marginal coverage. Under the stationarity and spatial mixing assumptions stated above, we instead establish asymptotic conditional coverage guarantees for LSCP. Proofs and additional technical details are deferred to Appendix A.

Our main theorem is the following, and With the tower law property, we can easily establish the same result (Corollary A.5) for marginal coverage of LSCP.

**Theorem 4.6** (Conditional coverage). *Under Assumption 4.1-4.3, for any $\alpha \in (0,1)$ and sample size $T$, we have*

$$\left| \mathbb{P}\left( Y(s_{n+1}) \in \widehat{C}_n\left(X(s_{n+1})\right) \mid X(s_{n+1}), s_{n+1} \right) - (1-\alpha) \right|$$
$$\leq 4L_{n+1}\delta_n + 6M_n n^{\frac{1+\gamma}{2}} + (2 + 4\sqrt{M}g(b))(\log_2 n + 2)^2 n^{-\gamma}. \tag{5}$$

*When $n \to \infty$, we have*

$$\mathbb{P}\left( Y(s_{n+1}) \in \widehat{C}_n\left(X(s_{n+1})\right) \mid X(s_{n+1}), s_{n+1} \right) \to 1 - \alpha. \tag{6}$$

From Inequality 5, we can see that the order of the coverage bound is controlled by $M_n n^{\frac{1+\gamma}{2}}$ and $n^{-\gamma}(\log_2 n + 2)^2$. The first term is equal to $n^{\frac{\gamma-1}{2}}$ in the special case of $M_n = \frac{1}{n}$ and the second term diminishes when $n$ is large enough because $n$ is of higher order than $\log_2 n$. As long as the estimation gap $\delta_n$ goes to zero when $n$ gets larger, the asymptotic conditional coverage can be inferred from the main theorem.

Table 2: Simulation: The table presents a comparison of the coverage and prediction interval width for five methods across three scenarios. The target coverage is 90%. The S1, S2 and S3 refers to Scenario 1,2,3.

| Method | S1 Coverage | S1 Width | S2 Coverage | S2 Width | S3 Coverage | S3 Width |
|--------|-------------|----------|-------------|----------|-------------|----------|
| LSCP | $0.902_{\pm 0.005}$ | $\mathbf{1.08}_{\pm 0.05}$ | $0.907_{\pm 0.006}$ | $\mathbf{0.53}_{\pm 0.03}$ | $0.914_{\pm 0.006}$ | $\mathbf{0.75}_{\pm 0.05}$ |
| EnbPI | $0.878_{\pm 0.005}$ | $1.39_{\pm 0.05}$ | $0.883_{\pm 0.007}$ | $0.64_{\pm 0.03}$ | $0.883_{\pm 0.006}$ | $0.93_{\pm 0.04}$ |
| GSCP | $0.905_{\pm 0.009}$ | $1.55_{\pm 0.06}$ | $0.915_{\pm 0.004}$ | $0.73_{\pm 0.04}$ | $0.916_{\pm 0.005}$ | $1.1_{\pm 0.05}$ |
| SLSCP | $0.91_{\pm 0.006}$ | $1.42_{\pm 0.05}$ | $0.903_{\pm 0.004}$ | $0.67_{\pm 0.03}$ | $0.916_{\pm 0.005}$ | $0.95_{\pm 0.05}$ |
| LCP | $0.902_{\pm 0.008}$ | $1.54_{\pm 0.05}$ | $0.915_{\pm 0.006}$ | $0.58_{\pm 0.03}$ | $0.914_{\pm 0.005}$ | $1.06_{\pm 0.05}$ |
| SLCP | $0.902_{\pm 0.008}$ | $1.33_{\pm 0.05}$ | $0.900_{\pm 0.006}$ | $0.64_{\pm 0.03}$ | $0.894_{\pm 0.007}$ | $0.874_{\pm 0.05}$ |

Table 3: Real data: The table presents a comparison of the coverage and prediction interval width for five methods on mobile signal data. The target coverage is 90%.

| Method | NM Coverage | NM Width | GA coverage | GA width |
|--------|-------------|----------|-------------|----------|
| LSCP | 0.926 | **211.3** | 0.906 | **130.8** |
| EnbPI | 0.884 | 272.2 | 0.886 | 166.3 |
| GSCP | 0.901 | 295.9 | 0.892 | 167.3 |
| SLSCP | 0.898 | 276.8 | 0.896 | 167.8 |
| LCP | 0.901 | 283.9 | 0.893 | 166.7 |
| SLCP | 0.896 | 266.2 | 0.899 | 166.5 |

## 5 EXPERIMENTS

In this section, we compare our proposed LSCP method with four baselines: EnbPI (Xu & Xie, 2021), which assigns equal weights to recent observations in time series; GSCP (Mao et al., 2024), which applies equal weights to all data points; SLSCP (Mao et al., 2024), which utilizes $k$-nearest neighbors and weights points based on spatial distance; and LCP (Guan, 2023), which weights all points according to feature similarity using a Gaussian kernel.

We randomly split the dataset into three subsets: 40% for training, 40% for calibration, and 20% for testing. For each method and dataset, the number of neighbors and the Gaussian kernel bandwidth are selected using 5-fold cross-validation on the calibration set. The prediction model we use in the experiment is the KNN regressor. More experiment details and choices of hyperparameters can be found in Appendix C.

### 5.1 SYNTHETIC DATA EXPERIMENTS

We begin by comparing LSCP with baseline methods across several simulated scenarios. For simplicity, we assume that the locations $s$ are uniformly sampled from the unit grid $[0,1] \times [0,1]$. The mean-zero stationary Gaussian process $X(s)$ is defined using a Matérn covariance function with variance $\sigma^2 = 1$, range $\phi = 0.1$, and smoothness $\kappa = 0.7$. The scenarios are as follows: (i) $Y(s) = X(s) + \epsilon(s)$, (ii) $Y(s) = X(s)|\epsilon(s)|$, and (iii) $Y(s) = X(s) + \sin(\|s\|_2)\epsilon(s)$.

These scenarios incorporate nonlinear and heteroskedastic settings that extend beyond the assumptions of our theoretical framework. Nevertheless, the empirical results demonstrate that LSCP consistently outperforms other methods across all scenarios.

In Table 2, LSCP consistently meets the nominal 90% coverage across all scenarios while delivering substantially narrower prediction intervals. Its small standard deviations (both for coverage and width) indicate that performance is stable. Figure 2 maps the prediction interval widths for different methods over space. We omit GSCP from these maps because it yields a constant width everywhere. Interpreting color as uncertainty and comparing to the true residual heatmap, LSCP clearly tracks sharp transitions and heteroskedastic pockets, whereas EnbPI and SLSCP show more homogenized patterns due to their local averaging.

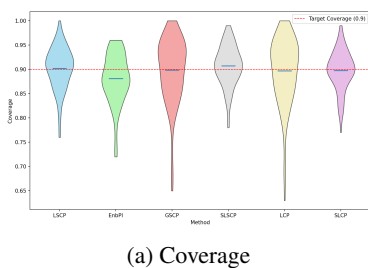 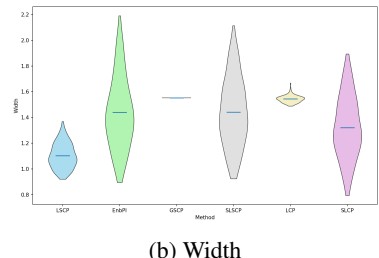

(a) Coverage
(b) Width

Figure 3: The violin plots on the left show the distribution of coverage across different areas for Scenario 1, while the plots on the right show the distribution of width.

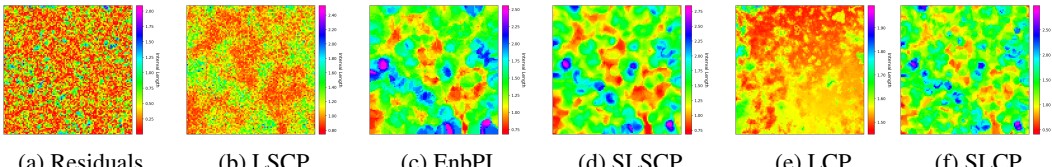

(a) Residuals    (b) LSCP    (c) EnbPI    (d) SLSCP    (e) LCP    (f) SLCP

Figure 4: The heatmaps illustrate the width of the prediction intervals for each method in Scenario 1. The width heatmap of LSCP closely matches the true residual heatmap, demonstrating its ability to capture fine details accurately.

To evaluate the spatial reliability of the methods, we further divide the grid into $10 \times 10$ areas and compute compute per-cell coverage and mean width, as shown in Figure 4. All methods except EnbPI achieve average coverage above the target $90\%$, with LSCP exhibiting the tightest dispersion and the smallest widths overall, indicating better use of data where uncertainty is low and appropriate widening where it is high. LCP behaves similarly to GSCP when the kernel bandwidth is large, since its weights flatten toward uniformity across the calibration set.

## 5.2 REAL DATA EXPERIMENTS

In the real-data experiments, we utilize mobile network measurement data from the Ookla public dataset, which records user-reported statewide mobile internet performance. Our analysis focuses on New Mexico and Georgia. The dataset is highly imbalanced between urban and suburban regions, creating substantial heterogeneity. Further details of the dataset can be found in Appendix C.3.

As shown in Table 3, our proposed LSCP method still outperforms competing methods by achieving significantly narrower prediction intervals while maintaining better coverage. Mirroring the synthetic study, we partition each state into $10 \times 10$ cells and compute per-cell coverage and mean width on the test set. The violin plots in Figure 4 expose spatial consistency rather than just global averages. Coverage is naturally higher in dense urban cells and lower in sparse rural regions. Notably, for New Mexico, the area-level coverage of all methods falls short of the nominal rate, whereas LSCP comes closest to the target. This shortfall is driven by severe data imbalance and sparsity in that state (Figure 1): some cells have too few informative neighbors, which depresses local coverage regardless of method. Even so, the locality and adaptive weighting in LSCP mitigate these effects better than the alternatives, yielding higher validity and lower interval tightness. The results show that LSCP consistently outperforms alternatives, achieving high coverage with narrow intervals. Violin plots confirm its spatial stability and robustness to non-uniform data distributions, corroborating our synthetic results and validating the method's effectiveness in complex settings.

## 6 CONCLUSION

We introduced LSCP, a localized spatial conformal method that learns adaptive weights from neighborhood residuals, avoiding hand-tuned kernels and restrictive infill assumptions. Empirically, LSCP yields tighter, consistent prediction intervals across synthetic and real datasets, demonstrating robustness and practicality. This framework naturally extends to spatio-temporal settings, where exchangeability fails but local structure persists.

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

## A    THEORETICAL DETAILS

### A.1    DERIVATION OF WEIGHTED QUANTILE

In our work, we employ the QRF from (Meinshausen & Ridgeway, 2006) to construct the quantile estimator. Within this framework, the estimated CDF can be expressed as a weighted empirical quantile.

Consider a feature vector $\tilde{X}_i$ with support $\mathrm{Supp}(\tilde{X}i) \subset \mathbb{B} \subset \mathbb{R}^p$. A regression tree $T(\theta)$ with parameter $\theta$ partitions $\mathbb{B}$ into $L$ disjoint rectangular subregions $R_l \subset \mathbb{B}$, each corresponding to a leaf node. By construction, these subregions cover $\mathbb{B}$, and every $x \in \mathbb{B}$ belongs to exactly one leaf, denoted $l(x, \theta)$. When $K$ trees are grown, each with its own parameter $\theta_k$, we define, for $x \in \mathbb{B}$ and observed features $\tilde{X}_1, \ldots, \tilde{X}_n$, the following weights:

$$k_\theta(l) := \#\{i \in \{1, \ldots, n\} : \tilde{X}_i \in R_{l(x,\theta)}\}, \tag{1}$$

$$w_i(x, \theta) := \frac{\mathbf{1}(\tilde{X}_i \in R_{l(x,\theta)})}{k_\theta(l)}, \tag{2}$$

$$w_i(x) := K^{-1} \sum_{k=1}^{K} w_i(x, \theta_k). \tag{3}$$

Here, (1) is the size of the leaf containing $x$, (2) assigns weight to the $i$-th observation according to its membership in this leaf normalized by leaf size, and (3) averages these weights across the $K$ trees. Using the aggregated weights, the estimated CDF is

$$\widehat{F}(y) := \sum_{i=1}^{n} w_i(x)\mathbf{1}(\tilde{Y}_i \le y). \tag{4}$$

Thus, $\widehat{F}(y)$ is a weighted empirical cumulative distribution function, where the weights are adaptively determined by the forest structure to reflect the proximity of each training point to the test input $x$ in feature space.

### A.2    PROOFS

We first remind the readers of the notations introduced in the paper. In Appendix A, we have shown that our method is equivalent to using a weighted empirical CDF as the estimate of the true CDF, where the weights $\omega_i$ are learnt from data. The weighted empirical distribution for the true noise $\varepsilon$ is

$$\tilde{F}_{n+1}(y) = \sum_{i=1}^{n} \omega_i \mathbf{1}(\varepsilon(s_i) \le y). \tag{5}$$

Here $\omega_n$ is the normalized weight. Besides, we also define the weighted empirical distribution for the residual $\hat{\varepsilon}$ as

$$\widehat{F}_{n+1}(y) = \sum_{i=1}^{n} \omega_i \mathbf{1}(\hat{\varepsilon}(s_i) \le y). \tag{6}$$

We assume an additive true model which is commonly used in literature like (Xu & Xie, 2021):

$$Y(s) = f(X(s)) + \varepsilon(s). \tag{7}$$

Considering that the residual is $\hat{\varepsilon}(s) = Y(s) - \hat{f}(X(s))$, it follows

$$\varepsilon(s) - \hat{\varepsilon}(s) = \hat{f}(X(s)) - f(X(s)). \tag{8}$$

The following Lemma bounds the distance between the weighted empirical distribution for the residual and true error.

**Lemma A.1** (Distance between the empirical CDF of $\varepsilon$ and $\hat{\varepsilon}$). *Under Assumption 4.1 and 4.2,*

$$\sup_y |\widehat{F}_{n+1}(y) - \widetilde{F}_{n+1}(y)| \leq (L_{n+1} + 1)\delta_n + 2\sup_y |\widetilde{F}_{n+1}(y) - F_\varepsilon(y)|. \tag{9}$$

*Proof.* Using Assumption 4.2, we have that

$$\sum_{i=1}^n \omega_i |\varepsilon(s_i) - \hat{\varepsilon}(s_i)| \leq M_n \sum_{i=1}^n |\varepsilon(s_i) - \hat{\varepsilon}(s_i)| \leq \delta_n^2. \tag{10}$$

Let $S = \{i : |\varepsilon(s_i) - \hat{\varepsilon}(s_i)| \geq \delta_n\}$. Then

$$\delta_n \sum_{i \in S} \omega_i \leq \sum_{i=1}^n \omega_i |\varepsilon(s_i) - \hat{\varepsilon}(s_i)| \leq \delta_n^2. \tag{11}$$

So $\sum_{i \in S} \omega_i \leq \delta_n$. Then

$$|\widehat{F}_{n+1}(y) - \widetilde{F}_{n+1}(y)| \leq \sum_{i=1}^n \omega_i |\mathbf{1}\{\hat{\varepsilon}(s_i) \leq y\} - \mathbf{1}\{\varepsilon(s_i) \leq y\}|$$

$$\leq \sum_{i \in S} \omega_i + \sum_{i \notin S} \omega_i |\mathbf{1}\{\hat{\varepsilon}(s_i) \leq y\} - \mathbf{1}\{\varepsilon(s_i) \leq y\}|$$

$$\overset{(i)}{\leq} \sum_{i \in S} \omega_i + \sum_{i \notin S} \omega_i \mathbf{1}\{|\varepsilon(s_i) - y| \leq \delta_n\}$$

$$\leq \sum_{i \in S} \omega_i + \sum_{i=1}^n \omega_i \mathbf{1}\{|\varepsilon(s_i) - y| \leq \delta_n\}$$

$$\leq \delta_n + \mathbb{P}(|\varepsilon(s_{n+1}) - y| \leq \delta_n) +$$

$$\sup_y \left| \sum_{i=1}^n \omega_i \mathbf{1}\{|\varepsilon(s_i) - y| \leq \delta_n\} - \mathbb{P}(|\varepsilon(s_{n+1}) - y| \leq \delta_n) \right|$$

$$= \delta_n + [F_\varepsilon(y + \delta_n) - F_\varepsilon(y - \delta_n)] + \sup_y \left| [\widetilde{F}_{n+1}(y + \delta_n) - \widetilde{F}_{n+1}(y - \delta_n)] \right.$$

$$\left. - [F_\varepsilon(y + \delta_n) - F_\varepsilon(y - \delta_n)] \right|$$

$$\overset{(ii)}{\leq} (L_{n+1} + 1)\delta_n + 2\sup_y |\widetilde{F}_{n+1}(y) - F_\varepsilon(y)|,$$

where $(i)$ is because $|\mathbf{1}\{a \leq y\} - \mathbf{1}\{b \leq y\}| \leq \mathbf{1}\{|b - y| \leq |a - b|\}$ for $a, b \in \mathbb{R}$ and $(ii)$ is because the Lipschitz continuity of $F_\varepsilon(y)$. $\qquad\square$

**Lemma A.2.** *Under assumption 4.1-4.3, we have*

$$E\left(\sup_y |\tilde{F}_{n+1}(y) - F_\varepsilon(y)|^2\right) \leq \frac{(2 + \log_2 n)^2}{n}(1 + 2\sqrt{M}g(b)). \tag{12}$$

*Proof.* Define $Z_{n+1}(y) = \tilde{F}_{n+1}(y) - F_\varepsilon(y)$. Besides, we also define $Z_{n+1}(A) = \sum_{i=1}^n \omega_i \mathbf{1}(X(s_i) \in A) - F_\varepsilon(\varepsilon \in A)$, where $A$ can be any region. Let $N$ be some positive integer to be chosen later. We first represent the CDF $F_\varepsilon(x)$ in base 2:

$$F_\varepsilon(x) = \sum_{i=1}^N b_i(x)2^{-i} + r_N(x), \tag{13}$$

where $r_N(x) \in [0, 2^{-N})$ and $b_i = 0$ or $1$.

For $l \in \{1, 2, \cdots, N\}$, define

$$B_l(x) = \sum_{i=1}^l b_i(x)2^{-i}. \tag{14}$$

We can define the points $x_i$ where $F_\varepsilon(x_i) = B_i(x)$. We have that

$$F_\varepsilon(x) - F_\varepsilon(x_i) = \sum_{i=l+1}^{N} a_i(x)2^{-i} + r_N(x) \le 2^{-l}. \tag{15}$$

As a result, we can partition $Z_{n+1}(x)$ into the following sum:

$$Z_{n+1}(F_\varepsilon^{-1}(x)) = Z_{n+1}(F_\varepsilon^{-1}(B_1(x))) + \sum_{i=1}^{N-1}(Z_{n+1}(F_\varepsilon^{-1}(B_{i+1}(x))) - Z_{n+1}(F_\varepsilon^{-1}(B_i(x))))$$
$$+ (Z_{n+1}(F_\varepsilon^{-1}(y)) - Z_{n+1}(F_\varepsilon^{-1}(B_N(x)))). \tag{16}$$

In order to bound $Z_{n+1}(y)$, we can bound each individual term instead. Since $B_{i+1}(x) - B_i(x) = b_{i+1}(x)2^{-(i+1)} \le 2^{-(i+1)}$, we know the interval $[B_i(x), B_{i+1}(x)]$ either has zero length, or it is equal to one region in the set $\{[(j-1)2^{-(i+1)}, j2^{-(i+1)}], 1 \le j \le 2^{i+1}\}$. As a result, we have

$$|Z_{n+1}(F_\varepsilon^{-1}(B_{i+1}(x))) - Z_{n+1}(F_\varepsilon^{-1}(B_i(x)))| \le$$
$$\sup_{j\in[1,2^{i+1}]} |Z_{n+1}(F_\varepsilon^{-1}(j2^{-(i+1)})) - Z_{n+1}(F_\varepsilon^{-1}((j-1)2^{-(i+1)}))| \tag{17}$$

Let

$$\delta_i = \sup_{j\in[1,2^{i+1}]} |Z_{n+1}([F_\varepsilon^{-1}(j2^{-(i+1)}), F_\varepsilon^{-1}((j-1)2^{-(i+1)})])|,$$

and

$$\delta_{xN} = \sup_x |Z_{n+1}([F_\varepsilon^{-1}(B_N(x)), x])|.$$

It follows that

$$|Z_{n+1}(F_\varepsilon^{-1}(y))| \le \sum_{i=1}^{N} \delta_i + \delta_{xN}. \tag{18}$$

By the triangle inequality,

$$(E(\sup_{y\in[0,1]} |Z_{n+1}(F_\varepsilon^{-1}(y))|^2))^{1/2} \le \sum_{i=1}^{N}(E\delta_i^2)^{1/2} + (E\delta_{xN}^2)^{1/2}. \tag{19}$$

Then we need to bound $\|\delta_i\|_2$ and $\|\delta_{xN}\|_2$ separately. Since $\delta_i$ is computing the supremum over a set, it can be bounded by the sum over the set,

$$\delta_i^2 \le \sum_{j=1}^{2^i}(Z_{n+1}(F_\varepsilon^{-1}(j2^{-(i+1)})) - Z_{n+1}(F_\varepsilon^{-1}((j-1)2^{-(i+1)}))^2. \tag{20}$$

Taking expectation, we have

$$E\delta_i^2 \le \sum_{j=1}^{2^i} E(Z_{n+1}(F_\varepsilon^{-1}(j2^{-(i+1)})) - Z_{n+1}(F_\varepsilon^{-1}((j-1)2^{-(i+1)}))^2$$
$$= \sum_{j=1}^{2^i} \text{Var}(Z_{n+1}([F_\varepsilon^{-1}((j-1)2^{-(i+1)}), F_\varepsilon^{-1}(j2^{-(i+1)})])). \tag{21}$$

Let $(\epsilon_i)_{i>0}$ be a sequence of independent and symmetric random variables in $\{-1, 1\}$. For any finite partition $A_1, \cdots, A_k$ of $\mathbb{R}$,

$$\sum_{j=1}^{k} \text{Var } Z_{n+1}(A_j) = E(Z_{n+1}^2(\sum_{j=1}^{k} \epsilon_i \mathbf{1}_{A_i}))$$
$$\overset{(i)}{\le} M_n^2(n + 2E \sum_{1\le i<j\le n} \alpha_{ij}), \tag{22}$$

where $\alpha_{ij} = \alpha(\sigma(\varepsilon(s_i)), \sigma(\varepsilon(s_j)))$ is the alpha-mixing coefficient, $M_n = \max_{1 \leq i \leq n} \omega_i$ and $(i)$ is because of Lemma 1.1 in (Rio et al., 2017).

Because of Assumption 4.3, we have

$$
E \sum_{1 \leq i < j \leq n} \alpha_{ij} \leq E \sum_{1 \leq i < j \leq n} \alpha_1(|s_i - s_j|)g(b)
$$

$$
\leq n \sqrt{E \sum_{1 \leq i < j \leq n} \alpha_1^2(|s_i - s_j|)g(b)}
$$

$$
\leq n^2 \sqrt{E_{d \sim g_n} \alpha_1^2(d)} g(b)
$$

$$
\leq n\sqrt{M}g(b), \tag{23}
$$

where $g_n$ is the distribution of the distance between $s_i$ and $s_j$ for any $i, j \in \{1, \cdots, n\}$.

Because $[F_\varepsilon^{-1}((j-1)2^{-(i+1)}), F_\varepsilon^{-1}(j2^{-(i+1)})]$ for $j = 1, \cdots, 2^{i+1}$ is a partition of $\mathbb{R}$, from 22,

$$
E\delta_i^2 \leq \sum_{j=1}^{2^i} \mathrm{Var}(Z_{n+1}([F_\varepsilon^{-1}((j-1)2^{-(i+1)}), F_\varepsilon^{-1}(j2^{-(i+1)})]))
$$

$$
\leq nM_n^2(1 + 2\sqrt{M}g(b)). \tag{24}
$$

For the other term $\delta_{xN}$, we know $x = F_\varepsilon^{-1}(F_\varepsilon(x)) \leq F_\varepsilon^{-1}(B_N(x) + r_N(x))$. We have

$$
Z_{n+1}([F_\varepsilon^{-1}(B_N(x)), x]) = \tilde{F}_{n+1}([F_\varepsilon^{-1}(B_N(x)), x]) - F_\varepsilon([F_\varepsilon^{-1}(B_N(x)), x])
$$

$$
\geq -F_\varepsilon([F_\varepsilon^{-1}(B_N(x)), x]) \tag{25}
$$

$$
= B_N(x) - x \geq -2^{-N}.
$$

On the other hand,

$$
Z_{n+1}([F_\varepsilon^{-1}(B_N(x)), x]) = Z_{n+1}([F_\varepsilon^{-1}(B_N(x)), B_N(x)) + 2^{-N}]) - Z_{n+1}([F_\varepsilon^{-1}(x, B_N(x)) + 2^{-N}])
$$

$$
\leq Z_{n+1}([F_\varepsilon^{-1}(B_N(x)), B_N(x)) + 2^{-N}]) + 2^{-N}. \tag{26}
$$

As a result, we have

$$
\delta_{xN} \leq \delta_N + 2^{-N}. \tag{27}
$$

To sum up, we prove that

$$
(E(\sup_{y \in [0,1]} |Z_{n+1}(F_\varepsilon^{-1}(y))|^2))^{\frac{1}{2}} \leq n^{\frac{1}{2}} M_n(N + 1 + 2^{-N})(1 + 2\sqrt{M}g(b))^{\frac{1}{2}}. \tag{28}
$$

Let $N = \log_2 n$, we have

$$
E(\sup_{y \in [0,1]} |Z_{n+1}(F_\varepsilon^{-1}(y))|^2) \leq M_n^2 n(2 + \log_2 n)^2(1 + 2\sqrt{M}g(b)). \tag{29}
$$

$\square$

**Lemma A.3** (Convergence of empirical CDF of $\varepsilon$). *Under Assumptions 4.1-4.3, with probability higher than $1 - (1 + 2\sqrt{M}g(b))(\log_2 n + 2)^2 n^{-\gamma}$,*

$$
\sup_y \left| \tilde{F}_{n+1}(y) - F_\epsilon(y) \right| \leq M_n n^{\frac{1+\gamma}{2}}. \tag{30}
$$

*Proof.* We have that

$$
\tilde{F}_{n+1}(y) - F_\epsilon(y) = \sum_{i=1}^n \omega_i \mathbf{1}(\epsilon(s_i) \leq y) - F_\epsilon(y)
$$

$$
= \sum_{i=1}^n \omega_i(\mathbf{1}(\epsilon(s_i) \leq y) - F_\epsilon(y)). \tag{31}
$$

Let $Z(s_i) = \mathbf{1}(\epsilon(s_i) \le y) - F_\epsilon(y)$, we know

$$EZ(s_i) = 0, \tag{32}$$

and $Z(s)$ is stationary. From Lemma A.2, using Markov inequality, we have that for any $k$,

$$\mathbb{P}(\sup_y |\tilde{F}_{n+1}(y) - F_\varepsilon(y)| \ge k) \le \frac{E(\sup_y |\tilde{F}_{n+1}(y) - F_\varepsilon(y)|^2)}{k^2}$$

$$= \frac{M_n^2 n(2 + \log_2 n)^2(1 + 2\sqrt{M}g(b))}{k^2}. \tag{33}$$

Let $k = M_n n^{\frac{1+\gamma}{2}}$, we have

$$\mathbb{P}\left(\sup_y |\tilde{F}_{n+1}(y) - F_\varepsilon(y)| \ge M_n n^{\frac{1+\gamma}{2}}\right) \le (2 + \log_2 n)^2(1 + 2\sqrt{M}g(b))n^{-\gamma}. \tag{34}$$

$\square$

**Theorem A.4.** *Under Assumption 4.1-4.3, for any $\alpha \in (0,1)$ and sample size $T$, we have*

$$\left|\mathbb{P}\left(Y(s_{n+1}) \in \widehat{C}_n\left(X(s_{n+1})\right) \mid X(s_{n+1}), s_{n+1}\right) - (1 - \alpha)\right|$$

$$\le 4L_{n+1}\delta_n + 6M_n n^{\frac{1+\gamma}{2}} + (2 + 4\sqrt{M}g(b))(\log_2 n + 2)^2 n^{-\gamma}. \tag{35}$$

*Proof.* For simplicity, we use $X_i = X(s_i)$, $Y_i = Y(s_i)$, $\varepsilon_i = \varepsilon(s_i)$, $\hat{\varepsilon}_i = \hat{\varepsilon}(s_i)$ and $\omega_{ni} = \omega_i$. For any $\beta \in [0,1]$,

$$\left|\mathbb{P}\left(Y_{n+1} \in \widehat{C}_n\left(X_{n+1}\right) \middle| X_{n+1}, s_{n+1}\right) - (1 - \alpha)\right|$$

$$= \left|\mathbb{P}\left(\hat{\varepsilon}_{n+1} \in [\widehat{Q}_\beta(X_{n+1}), \widehat{Q}_{1-\alpha+\beta}(X_{n+1})] \middle| X_{n+1}, s_{n+1}\right) - (1 - \alpha)\right|$$

$$= \left|\mathbb{P}\left(\hat{\varepsilon}_{n+1} \in [\widehat{Q}_\beta(X_{n+1}), \widehat{Q}_{1-\alpha+\beta}(X_{n+1})] \middle| X_{n+1}\right) - (1 - \alpha)\right|$$

$$= \left|\mathbb{P}\left(\beta \le \sum_{i=1}^n \omega_{ni}\mathbf{1}(\hat{\varepsilon}_i \le \hat{\varepsilon}_{n+1}) \le 1 - \alpha + \beta \middle| X_{n+1}\right) - (1 - \alpha)\right|$$

$$= \left|\mathbb{P}\left(\beta \le \widehat{F}_{n+1}(\hat{\varepsilon}_{n+1}) \le 1 - \alpha + \beta \middle| X_{n+1}\right) - \mathbb{P}(\beta \le F_\varepsilon(\varepsilon_{n+1}) \le 1 - \alpha + \beta)\right|$$

$$= \left|\mathbb{P}\left(\beta \le \widehat{F}_{n+1}(\hat{\varepsilon}_{n+1}) \le 1 - \alpha + \beta \middle| X_{n+1}\right) - \mathbb{P}\left(\beta \le F_\varepsilon(\varepsilon_{n+1}) \le 1 - \alpha + \beta \middle| X_{n+1}\right)\right|$$

$$\le \mathbb{E}\left(\left|\mathbf{1}\{\beta \le \widehat{F}_{n+1}(\hat{\varepsilon}_{n+1}) \le 1 - \alpha + \beta\} - \mathbf{1}\{\beta \le F_\varepsilon(\varepsilon_{n+1}) \le 1 - \alpha + \beta\}\right| \middle| X_{n+1}\right)$$

$$\stackrel{(i)}{\le} \mathbb{E}\left(\left|\mathbf{1}\{\beta \le \widehat{F}_{n+1}(\hat{\varepsilon}_{n+1})\} - \mathbf{1}\{\beta \le F_\varepsilon(\varepsilon_{n+1})\}\right|\right.$$

$$\left. + \left|\mathbf{1}\{\widehat{F}_{n+1}(\hat{\varepsilon}_{n+1}) \le 1 - \alpha + \beta\} - \mathbf{1}\{F_\varepsilon(\varepsilon_{n+1}) \le 1 - \alpha + \beta\}\right| \middle| X_{n+1}\right)$$

$$\stackrel{(ii)}{\le} \mathbb{P}\left(|F_\varepsilon(\varepsilon_{n+1}) - \beta| \le \left|F_\varepsilon(\varepsilon_{n+1}) - \widehat{F}_{n+1}(\hat{\varepsilon}_{n+1})\right| \middle| X_{n+1}\right)$$

$$+ \mathbb{P}\left(|F_\varepsilon(\varepsilon_{n+1}) - (1 - \alpha + \beta)| \le \left|F_\varepsilon(\varepsilon_{n+1}) - \widehat{F}_{n+1}(\hat{\varepsilon}_{n+1})\right| \middle| X_{n+1}\right),$$

where Inequality $(i)$ follows since for any constants $a, b$ and univariates $x, y$, $|\mathbf{1}\{a \le x \le b\} - \mathbf{1}\{a \le y \le b\}| \le |\mathbf{1}\{a \le x\} - \mathbf{1}\{a \le y\}| + |\mathbf{1}\{x \le b\} - \mathbf{1}\{y \le b\}|$. On the other hand, Inequality $(ii)$ is a result of $|\mathbf{1}\{a \le x\} - \mathbf{1}\{b \le x\}| \le \mathbf{1}\{|b - x| \le |a - b|\}$.

Using Lemma A.2,

$$\mathbb{P}\left(\left|F_\varepsilon(\varepsilon_{n+1}) - \beta\right| \leq \left|F_\varepsilon(\varepsilon_{n+1}) - \widehat{F}_{n+1}(\hat{\varepsilon}_{n+1})\right| \,\middle|\, X_{n+1}\right)$$

$$\leq \mathbb{P}\left(\left|F_\varepsilon(\varepsilon_{n+1}) - \beta\right| \leq \left|F_\varepsilon(\varepsilon_{n+1}) - \widehat{F}_{n+1}(\hat{\varepsilon}_{n+1})\right|, \sup_y \left|F_\varepsilon(y) - \widehat{F}_{n+1}(y)\right| \leq M_n n^{\frac{1+\gamma}{2}} \,\middle|\, X_{n+1}\right)$$

$$+ \mathbb{P}\left(\sup_y \left|F_\varepsilon(y) - \widehat{F}_{n+1}(y)\right| \geq M_n n^{\frac{1+\gamma}{2}} \,\middle|\, X_{n+1}\right)$$

$$\leq \mathbb{P}\left(\left|F_\varepsilon(\varepsilon_{n+1}) - \beta\right| \leq \left|F_\varepsilon(\varepsilon_{n+1}) - \widehat{F}_{n+1}(\hat{\varepsilon}_{n+1})\right| \,\middle|\, \sup_y \left|F_\varepsilon(y) - \widehat{F}_{n+1}(y)\right| \leq M_n n^{\frac{1+\gamma}{2}}, X_{n+1}\right)$$

$$+ \mathbb{P}\left(\sup_y \left|F_\varepsilon(y) - \widehat{F}_{n+1}(y)\right| \geq M_n n^{\frac{1+\gamma}{2}}\right)$$

$$\leq \mathbb{P}\left(\left|F_\varepsilon(\varepsilon_{n+1}) - \beta\right| \leq |F_\varepsilon(\varepsilon_{n+1}) - F_\varepsilon(\hat{\varepsilon}_{n+1})| + (L_{n+1}+1)\delta_n + 3M_n n^{\frac{1+\gamma}{2}} \,\middle|\, X_{n+1}\right)$$

$$+ (1 + 2\sqrt{M}g(b))(\log_2 n + 2)^2 n^{-\gamma}$$

$$\leq \mathbb{P}\left(\left|F_\varepsilon(\varepsilon_{n+1}) - \beta\right| \leq L_{n+1}|\varepsilon_{n+1} - \hat{\varepsilon}_{n+1}| + (L_{n+1}+1)\delta_n + 3M_n n^{\frac{1+\gamma}{2}} \,\middle|\, X_{n+1}\right)$$

$$+ (1 + 2\sqrt{M}g(b))(\log_2 n + 2)^2 n^{-\gamma}$$

$$\leq 2L_{n+1}\delta_n + 3M_n n^{\frac{1+\gamma}{2}} + (1 + 2\sqrt{M}g(b))(\log_2 n + 2)^2 n^{-\gamma}.$$

The inequality above also holds for $\mathbb{P}\left(\left|F_\varepsilon(\varepsilon_{n+1}) - \beta\right| \leq |F_\varepsilon(\varepsilon_{n+1}) - \widehat{F}_{n+1}(\hat{\varepsilon}_{n+1})| \,\middle|\, X_{n+1}\right)$, and we can conclude that

$$\left|\mathbb{P}\left(Y_{n+1} \in \widehat{C}_n\left(X_{n+1}\right) \mid X_{n+1}, s_{n+1}\right) - (1-\alpha)\right| \tag{36}$$
$$\leq 4L_{n+1}\delta_n + 6M_n n^{\frac{1+\gamma}{2}} + (2 + 4\sqrt{M}g(b))(\log_2 n + 2)^2 n^{-\gamma}.$$

$\square$

**Corollary A.5** (Marginal Coverage). *Under Assumption 4.1-4.3, for any $\alpha \in (0,1)$ and sample size T, we have*

$$\left|\mathbb{P}\left(Y(s_{n+1}) \in \widehat{C}_n\left(X(s_{n+1})\right)\right) - (1-\alpha)\right| \tag{37}$$
$$\leq 4L_{n+1}\delta_n + 6M_n n^{\frac{1+\gamma}{2}} + (2 + 4\sqrt{M}g(b))(\log_2 n + 2)^2 n^{-\gamma}.$$

## B    COMPARISON WITH OTHER METHODS

**LSCP vs. GSCP**    Global Spatial Conformal Prediction (GSCP), introduced in (Mao et al., 2024), applies equal weighting to all non-conformity scores across the calibration dataset, and the estimated quantile at any point is given by

$$\widehat{Q}_n(p) = \inf\{e \in \mathbb{R} : \frac{1}{n}\sum_{i=1}^n \mathbf{1}\{\hat{\varepsilon}(s_i) \leq e\} \leq p\}.$$

This approach can be viewed as an extension of split conformal prediction and its key strength lies in the coverage guarantee obtained under minimal assumptions. In particular, when spatial locations are sampled i.i.d. from a common distribution, the data are exchangeable, as permuting the order does not alter the joint probability law. In this case, GSCP naturally ensures valid marginal coverage.

In practice, however, GSCP is often too conservative because real-world data distributions vary across locations. The method enforces uniform weights on all the training data, and captures local adaptivity only through a user-specified variance estimate $\hat{\sigma}$ in the non-conformity score. As a result, GSCP

intervals are typically achieving coverage by being unnecessarily wide in some regions. A natural remedy is to localize the procedure by estimating empirical quantiles within neighborhoods, but doing so breaks the spatial exchangeability, and thus the theoretical guarantee no longer holds.

**LSCP vs. SLSCP**    Smoothed Local Spatial Conformal Prediction (SLSCP) is another method from (Mao et al., 2024) that improves GSCP by using only nearby data to construct prediction intervals, with the weighted empirical quantile estimated locally as:

$$\widehat{Q}_n(p) = \inf\{e \in \mathbb{R} : \sum_{i \in N(s_{n+1})} \omega_i \mathbf{1}\{\hat{\varepsilon}(s_i) \leq e\} \leq p\}.$$

SLSCP assigns weights $\omega_i \propto k(\|s_i - s_{n+1}\|)$, relying solely on spatial distance. In contrast, our method learns data-adaptive $\omega_i$ by training quantile regression with features $\tilde{X}(s)$, capturing richer information beyond distance. As shown in Section 5, this makes our approach more adaptive and significantly improves performance over SLSCP.

Our theoretical framework also differs fundamentally from theirs. SLSCP relies on infill sampling assumption, requiring data points to become arbitrarily dense around $s_{n+1}$, and assumes a spatially continuous process with locally i.i.d. noise. These conditions yield local asymptotic exchangeability, which lead to coverage guarantee. By contrast, our approach assumes only a stationary, spatially mixing error process, our method avoids the need for infinitely close neighbors and generalizes more broadly. Moreover, this framework naturally extends to spatio-temporal settings, making LSCP applicable in real-world scenarios where SLSCP fails—particularly for time series data, which violate the infill assumption.

**LSCP vs. LCP**    Localized Conformal Prediction (LCP), introduced in (Guan, 2023), provides a general framework for localized conformal prediction rather than the spatial setting only. The method combines GSCP and SLSCP in quantile estimation:

$$\widehat{Q}_n(p) = \inf\{e \in \mathbb{R} : \sum_{i=1}^{n} \omega_i \mathbf{1}\{\hat{\varepsilon}(s_i) \leq e\} \leq p\}.$$

Similar to GSCP, LCP uses all calibration data for prediction, but like SLSCP, it applies different weights to each data point, which are defined as $\omega_i \propto k(X(s_i), X(s_{n+1}))$, where $k$ is a user-specified kernel function. While this kernel-based design provides more flexibility than purely location-based weights in SLSCP, it has two key limitations: (i) it only encodes pairwise similarity between features, and (ii) the kernel must be hand-specified by the user, which makes performance highly sensitive to this choice.

Our proposed LSCP removes these limitations by directly learning data-adaptive weights. Instead of relying on a user-specified kernel, LSCP uses quantile regression on neighborhood residuals to infer weights automatically from the data. This not only captures richer local dependencies that fixed kernels cannot, but also makes the method far easier to apply in practice, since it eliminates the need to tune or select a kernel function. As a result, LSCP is both more expressive and more user-friendly than LCP, while achieving stronger empirical performance.

Another limitation of LCP lies in its theoretical assumptions: it requires the data $(X_i, Y_i)_{i=1}^{n}$ to be i.i.d. to guarantee finite-sample marginal coverage. This assumption is stronger than those of other conformal methods mentioned, and it restricts the generality and applicability of the results.

## C    EXPERIMENTAL DETAILS

### C.1    BASELINE METHODS

In our experiments, we compare LSCP against four baselines spanning time series and spatial conformal prediction:

1. **EnbPI** (Xu & Xie, 2021): a general framework for time series prediction intervals that fits leave-one-out regressors and uses their residuals as nonconformity scores. Quantiles are computed from the empirical (uniformly weighted) residual distribution from a past window.

2. **GSCP** (Mao et al., 2024): a spatial CP method that extends split conformal to spatial settings by using a global empirical quantile with uniform weights over all calibration points.

3. **SLSCP** (Mao et al., 2024): a localized spatial CP method that estimates a weighted quantile using a distance-based kernel, thereby adapting to local spatial structure.

4. **LCP** (Guan, 2023): a general localized CP framework that assigns weights to all calibration points via a user-specified kernel on feature similarity, aiming to capture feature-level dependence.

## C.2 DATA SIMULATION

In order to compare the performance of different methods, we simulate several different scenarios that go beyond the assumptions made in the paper.

We uniformly sample the spatial locations $s$ from the unit grid $[0, 1] \times [0, 1]$. For each scenario, we sample 12000 data points, where $40\%$ are used as training data, $40\%$ as calibration data and the rest $20\%$ as test data. Let $X(s)$ be a mean-zero stationary Gaussian process with Matérn covariance (variance $\sigma^2 = 1$, range $\phi = 0.1$, smoothness $\kappa = 0.7$). We use the three regimes that have been considered in spatial CP literature:

1. Stationary, homoskedastic spatial data:

$$Y(s) = X(s) + \varepsilon(s).$$

This scenario establishes a reference case close to standard assumptions (stationary signal with additive, location-invariant noise), ensuring methods behave sensibly when global structure suffices.

2. Heteroskedastic spatial data:

$$Y(s) = X(s)|\varepsilon(s)|.$$

This scenario introduces heteroskedasticity tied to the latent field (variance depends on $|X(s)|$) and non-Gaussian noise, stresses robustness to model misspecification and the ability to adapt via local residual information.

3. Heteroskedastic and Non-stationary spatial data:

$$Y(s) = X(s) + \sin(\|s\|_2)\varepsilon(s).$$

This scenario creates spatially varying uncertainty (larger near peaks of $\sin(\|s\|_2)$, smaller near zeros), violates stationarity and highlights the benefit of localization that adapts to spatial heterogeneity.

## C.3 REAL-DATA DESCRIPTION

The dataset utilized in our study is sourced from the open datasets provided by Ookla. The data gathered by Ookla from 2019 to 2023 encapsulates the performance metrics of mobile internet connections for a multitude of users worldwide. Key variables in this dataset include geographical coordinates (longitude and latitude), mean download speed (MB/s), mean upload speed (MB/s), count of tests conducted in each area (aggregated for user privacy into $600m^2$ grid blocks), the number of distinct devices utilized for testing, and a comprehensive speed score assessing the connection speed. In our experiment, we utilize mobile connection data from the states of Georgia in the United States Southeast and New Mexico in the United States Southwest. In our experiments, we use mobile connection data from Georgia and New Mexico (USA) and aim to predict the speed score at new spatial locations. The New Mexico dataset contains $24,983$ observations, while the Georgia dataset includes $28,587$ data points.

## C.4 HYPERPARAMETER CHOICE

In our experiments we fix the base predictor to a k-nearest-neighbors regressor with $k = 5$, where $k$ is chosen by 5-fold cross validation. This keeps the mean model simple and transparent while letting us focus on how different uncertainty wrappers behave. In principle, any reasonable prediction model could replace KNN without changing the conformal machinery.

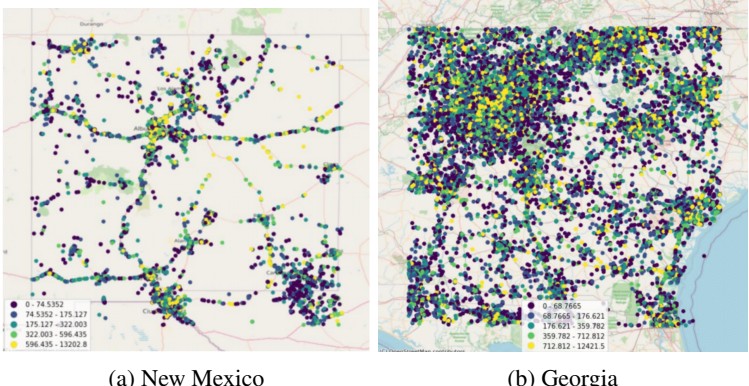

(a) New Mexico  (b) Georgia

Figure 1: Mobile signal score from Ookla dataset. The plots show the distribution of data accross the state.

For locality-based methods (LSCP, EnbPI and SLSCP), we use neighborhoods of 50 points. Empirically, coverage and width are fairly insensitive to this choice once the neighborhood is large enough to yield stable empirical quantiles but not so large that it washes out spatial heterogeneity.

For the kernel-smoothed methods (SLSCP and LCP), We tune it over $[0, 1]$ via five-fold cross-validation, selecting the smallest bandwidth that achieves nominal coverage on held-out folds while minimizing interval width. If no choice achieves the nominal coverage, then we choose the one that generates the closest coverage to it. The choice is $0.03$ for SLSCP and $0.5$ for LCP.

Finally, for quantile model in LSCP we use a Quantile Random Forest with $50$ trees and maximum depth $10$. Deeper trees tend to produce slightly tighter intervals by capturing finer structure, but at the cost of increased computation and a higher risk of overfitting in small local neighborhoods.

## C.5 COMPUTATIONAL COMPLEXITY

A potential concern with localized methods is the computational cost associated with fitting models on large spatial datasets. It is important to clarify that LSCP does *not* require training a separate model for each test location. Instead, the method maintains scalability through a two-stage design: (i) training a **single, global** Quantile Random Forest (QRF) on the residual-feature pairs, and (ii) performing lightweight, local information retrieval at inference time.

**Training Complexity.** The training phase consists of fitting one QRF model on the calibration residuals $\{(\tilde{X}(s_i), \tilde{Y}(s_i))\}_{i=1}^n$. The computational complexity is $\mathcal{O}(T \cdot n \log n)$, where $n$ is the sample size and $T$ is the number of trees. This cost is identical to fitting a standard Random Forest and scales efficiently to datasets with millions of points using parallelized implementations.

**Inference Complexity.** For a new test location $s_{n+1}$, the computational cost involves two steps:

1. **Neighbor Search:** Retrieving the $k$-nearest spatial neighbors. Using spatial indexing structures such as Ball Trees or $k$-d trees, this query can be performed in $\mathcal{O}(k \log n)$ time.

2. **Model Evaluation:** A single forward pass of the pre-trained QRF. Since the input dimension is determined by the neighborhood size $k$ (which is fixed and small), this step is $\mathcal{O}(T \cdot d_{\max})$, where $d_{\max}$ is the maximum depth of the trees.

Because the input dimension to the QRF depends only on $k$ and not on the total dataset size $n$, the inference cost remains low even as the spatial domain grows. Consequently, the total complexity of LSCP is comparable to standard global conformal prediction pipelines, making it tractable for massive spatial datasets.

## C.6 SENSITIVITY ANALYSIS

We investigate the sensitivity of the LSCP method to the choice of the neighborhood size $k$, which determines the locality of the weighted conformal prediction. Table 1 reports the empirical coverage and average interval width across a wide range of values, $k \in \{5, \ldots, 150\}$, for both the New Mexico and Georgia datasets. The results demonstrate that the proposed method is highly robust to the selection of $k$. The empirical coverage remains stable and consistently above the nominal level (ranging from 0.923 to 0.926 for NM and 0.905 to 0.909 for GA). Similarly, the average interval width exhibits only minor fluctuations, increasing slightly as $k$ grows large but maintaining efficiency compared to other baseline methods. This stability suggests that LSCP does not require precise hyperparameter tuning to achieve valid and efficient inference.

| $k$ | NM | | GA | |
|---|---|---|---|---|
| | Coverage | Width | Coverage | Width |
| 5 | 0.923 | 208.40 | 0.908 | 129.29 |
| 8 | 0.923 | 211.63 | 0.908 | 130.17 |
| 10 | 0.925 | 212.56 | 0.908 | 130.95 |
| 15 | 0.924 | 213.22 | 0.907 | 132.67 |
| 20 | 0.926 | 213.41 | 0.905 | 131.53 |
| 30 | 0.924 | 213.33 | 0.909 | 132.49 |
| 50 | 0.925 | 216.25 | 0.909 | 132.86 |
| 75 | 0.924 | 216.36 | 0.909 | 133.05 |
| 100 | 0.924 | 217.96 | 0.906 | 133.13 |
| 150 | 0.923 | 219.94 | 0.907 | 133.97 |

Table 1: Sensitivity of LSCP coverage and average interval width to the neighborhood size $k$ on the NM and GA datasets.

## C.7 ADDITIONAL REAL-WORLD EXPERIMENTS

To further demonstrate the robustness and efficiency of LSCP, we evaluate its performance on three additional real-world datasets covering agricultural and real estate domains.

**Crop Yield Prediction.** We first analyze a farming dataset that captures environmental and operational variables affecting crop yield across 500 farms situated in diverse geographical regions, including India, the USA, and Africa. As shown in Table 2, LSCP achieves the target coverage of 0.90 while producing the narrowest average prediction intervals (3605.33) compared to baselines such as EnbPI and GSCP.

**Real Estate Price Prediction.** We further assess the method on two large-scale housing datasets. The first is the California Housing dataset, comprising $20,640$ census data points. The second is the King County House Sales dataset, containing $21,613$ transactions. The results, presented in Tables 2, follow the same trend: LSCP maintains valid empirical coverage (approx. 0.90) while consistently delivering the tightest prediction intervals among all compared methods.

Table 2: Performance comparison (Coverage and Average Width) on three real-world datasets: Farming ($N = 500$), California Housing ($N = 20,640$), and King County House Sales ($N = 21,613$).

| Method | Farming | | California Housing | | King County | |
|---|---|---|---|---|---|---|
| | Coverage | Width | Coverage | Width | Coverage | Width |
| LSCP | 0.900 | **3,605.33** | 0.898 | **159,370.08** | 0.903 | **383,316.34** |
| EnbPI | 0.910 | 3,686.90 | 0.894 | 165,334.41 | 0.906 | 407,719.42 |
| GSCP | 0.910 | 3,687.25 | 0.895 | 196,959.02 | 0.903 | 419,219.85 |
| SLSCP | 0.900 | 3,694.17 | 0.893 | 164,571.93 | 0.910 | 401,277.29 |
| LCP | 0.900 | 3,702.04 | 0.899 | 197,504.70 | 0.904 | 420,069.28 |
| SLCP | 0.900 | 3,675.05 | 0.896 | 193,498.63 | 0.903 | 419,217.46 |

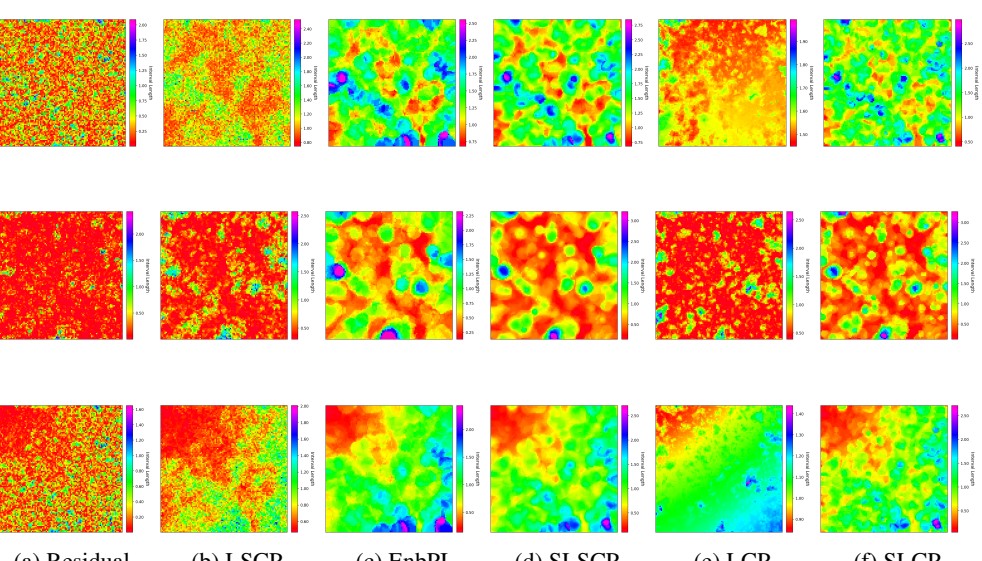

| (a) Residual | (b) LSCP | (c) EnbPI | (d) SLSCP | (e) LCP | (f) SLCP |
|---|---|---|---|---|---|

Figure 2: The heatmaps illustrate the width of the prediction intervals for each method across the three scenarios. The width heatmap of LSCP closely matches the true residual heatmap, demonstrating its ability to capture fine details accurately.

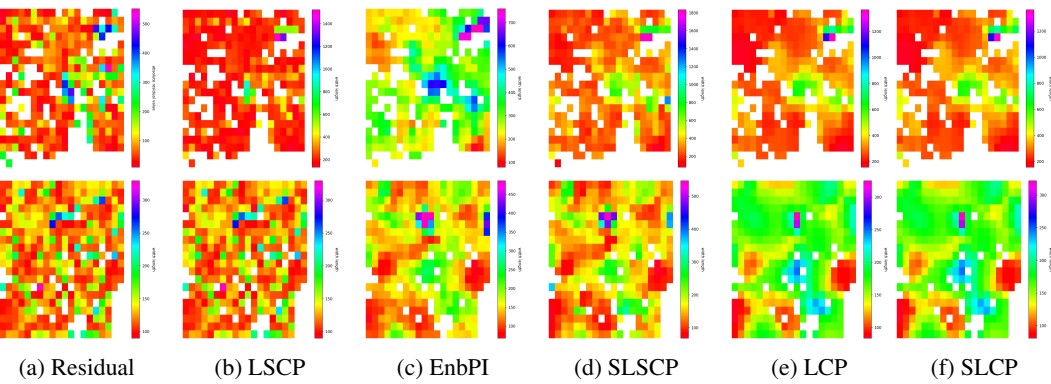

| (a) Residual | (b) LSCP | (c) EnbPI | (d) SLSCP | (e) LCP | (f) SLCP |
|---|---|---|---|---|---|

Figure 3: The heatmaps illustrate the width of the prediction intervals for each method in the NM (top row) and GA (bottom row) datasets.

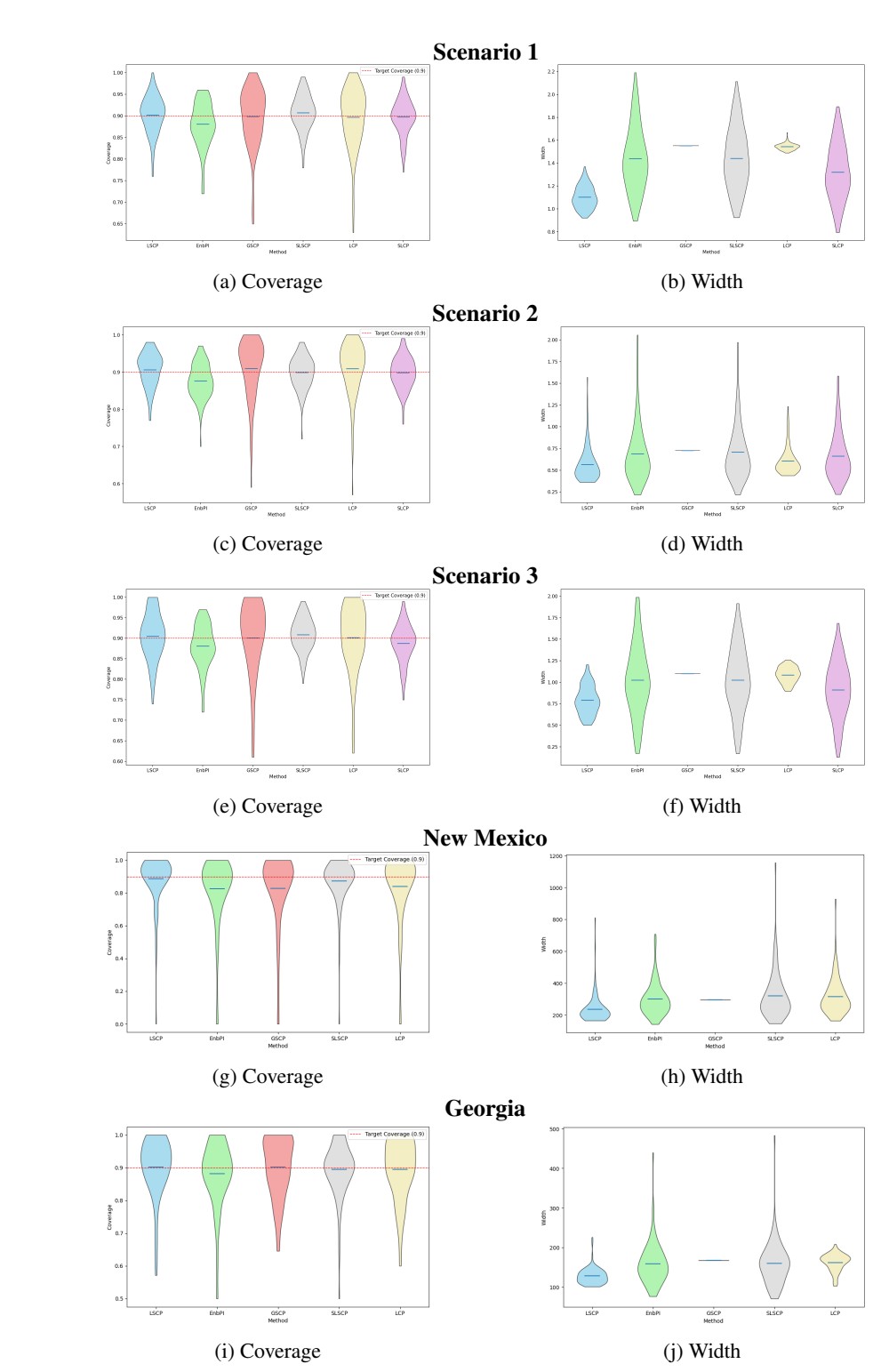

Figure 4: The violin plots on the left show the distribution of coverage across different areas, while the plots on the right show the distribution of width. Each row represents a different scenario or location.

