# OpenReview forum: "SPATIAL CONFORMAL INFERENCE THROUGH LOCALIZED QUANTILE REGRESSION"
_ICLR.cc/2026/Conference — Submitted to ICLR 2026_

### Official Review · Reviewer_9cdB · 2025-10-29

**Soundness:** 3
**Presentation:** 3
**Contribution:** 2
**Rating:** 4
**Confidence:** 4

**Summary:**

This paper addresses uncertainty quantification for spatial prediction, where traditional methods like Kriging require strong parametric assumptions that often fail in complex datasets, while standard conformal prediction ignores spatial dependence. The authors propose Localized Spatial Conformal Prediction (LSCP), a model-agnostic framework that learns data-adaptive weights through quantile regression on neighborhood residuals to construct spatially adaptive prediction intervals. LSCP automatically learns optimal local weights rather than relying on hand-tuned kernels, capturing complex spatial dependencies. Theoretically, the paper establishes finite-sample marginal coverage under exchangeability and proves asymptotic conditional coverage under the weaker assumptions of stationarity and spatial mixing, with explicit convergence rates. Empirically, experiments on synthetic and real datasets demonstrate that LSCP achieves near-nominal coverage with tighter and more stable intervals than existing conformal prediction methods.

**Strengths:**

1. The paper demonstrates that local residual patterns contain rich information about spatial uncertainty structure, which can be learned rather than manually specified. This shift from user-designed kernels to data-driven weight learning represents a paradigm change applicable beyond spatial settings.

2. Separating the base predictor from the uncertainty quantifier makes the method model-agnostic and easily deployable.

**Weaknesses:**

1. While the authors mention spatio-temporal extensions, the current framework is static. Many spatial applications (climate modeling, traffic prediction) have inherent temporal dependencies that aren't captured.

2. The paper doesn't address computational complexity for massive datasets. The QRF-based approach requires fitting models on neighborhood residuals, which could be prohibitive for millions of spatial locations.

3. The core idea of using data-adaptive weights isn't entirely novel: it extends existing weighted CP frameworks (Tibshirani et al., 2019; Barber et al., 2023) by learning weights through QRF rather than specifying them. The "localization" concept has also been introduced, the authors seem to miss another potential baseline here: Han et al., (2022) Split Localized Conformal Prediction.

4. The proof techniques largely follow standard approaches from the time series CP literature, adapting them to spatial settings through mixing conditions.

5. Limited diversity in real datasets, where only mobile signal data is tested.

**Questions:**

The k-nearest neighbor approach can produce artifacts at spatial boundaries or in regions with highly irregular sampling density, is this ever observed in the experiment and how it might affect the LSCP results?

---

> ### Author Response · Authors · 2025-11-20
>
> We sincerely thank the reviewer for the constructive and thoughtful feedback. We have uploaded a new version which includes more experiments and computational complexity analysis. We would be grateful if the reviewer could review the revised version. Below, we provide our point-by-point responses to the raised concerns and questions.
>
> > Weakness 1: While the authors mention spatio-temporal extensions, the current framework is static. Many spatial applications (climate modeling, traffic prediction) have inherent temporal dependencies that aren't captured.
>
> We thank the reviewer for raising the concern about spatio-temporal extensions.  While our theoretical development focuses on the static spatial setting, the framework naturally generalizes to spatio-temporal data. In particular, the temporal index can be incorporated into the spatial location by augmenting the coordinate with time, effectively treating $(s,t)$ as a point in a higher-dimensional space. Under this view, our assumptions remain meaningful: spatio-temporal mixing requires that correlations decay as observations become farther apart either in space or in time.
>
> At any prediction time $t$, one can capture both temporal and spatial dependence by defining the neighborhood as the $k$-nearest neighbors (in spatial location) within a recent time window (e.g., the past $m$ observations), and then applying our localized conformal prediction framework exactly as in the static case. This extension preserves the locality structure and weighting mechanism while allowing the method to adapt to dynamic environments.
>
> > Weakness 2: The paper doesn't address computational complexity for massive datasets. The QRF-based approach requires fitting models on neighborhood residuals, which could be prohibitive for millions of spatial locations.
>
> We appreciate the reviewer’s concern regarding computational scalability and we have made this clear in the appendix of our revised version. In short, our method remains computationally efficient for large datasets because
>
> (i) LSCP trains only a single QRF model, not one model per test location,
>
> (ii) nearest-neighbor search is lightweight and highly scalable.
>
> More specifically, LSCP does not train a QRF for each spatial location. Instead, we train one Quantile Random Forest on the calibration residual--feature pairs $\{(\tilde X(s_i),\tilde Y(s_i))\}_{i=1}^n$, which has computational complexity $O(Tn\log n)$ (where $T$ is the number of trees) essentially identical to fitting a standard random forest on $n$ samples. At prediction time, each test location only requires:
>
> (i) a $k$-nearest-neighbor query in the spatial domain, and
>
> (ii) a single forward pass of the pre-trained QRF on the extracted neighbor-residual feature vector.
>
> Both steps are extremely efficient. kNN queries can be accelerated to nearly logarithmic time, and QRF evaluation is $O(T)$ where $T$ is the number of trees.
>
> Moreover, the neighborhood size $k$ is very small (e.g., $k=5$--$20$), so the dimension of the QRF input is fixed and does not scale with the dataset size. This design choice ensures that LSCP remains computationally tractable even when the number of spatial locations reaches millions.
>
> > Weakness 3 part 1: The core idea of using data-adaptive weights isn't entirely novel: it extends existing weighted CP frameworks (Tibshirani et al., 2019; Barber et al., 2023) by learning weights through QRF rather than specifying them.
>
> We thank the reviewer for pointing out these related frameworks. In short, our method is fundamentally different from prior weighted or localized CP methods because it **learns data-adaptive spatial weights** from residual geometry, rather than manually specifying them or relying on pre-defined feature distances.
>
> This enables LSCP to capture complex spatial structure and heteroskedasticity that existing approaches cannot adapt to. The ability of the method to learn weights from the residual neighborhood leads to the efficiency and robustness of our method in the experiment.
>
> > Weakness 3 part 2: The "localization" concept has also been introduced, the authors seem to miss another potential baseline here: Han et al., (2022) Split Localized Conformal Prediction.
>
> We thank the reviewer for suggesting SLCP as an additional baseline. We have incorporated SLCP into our updated experiments and found that our method still consistently yields narrower prediction intervals while maintaining nominal coverage. Here is the result on Mobile Signal dataset:
>
> | Method | Width (GA) | Cov (GA) | Width (NM) | Cov (NM) |
> |--------|-----------:|----------:|-----------:|----------:|
> | LSCP   |    130.8   |   0.90   |    211.3   |   0.92   |
> | EnbPI  |    166.3   |   0.88   |    272.2   |   0.88   |
> | GSCP   |    167.3   |   0.89   |    295.9   |   0.90   |
> | SLSCP  |    167.3   |   0.89   |    276.8   |   0.89   |
> | LCP    |    167.8   |   0.89   |    283.9   |   0.90   |
> | SLCP   |    166.5   |   0.90   |    266.2   |   0.89   |

---

> ### Author Response · Authors · 2025-11-20
>
> > Weakness 4: The proof techniques largely follow standard approaches from the time series CP literature, adapting them to spatial settings through mixing conditions.
>
> We thank the reviewer for raising such concern. We acknowledge that our proof strategy draws inspiration from classical arguments in time-series conformal prediction. However, our work represents the first application of mixing conditions to the spatial conformal prediction setting—an extension that necessitates substantial technical innovation. Unlike time-series methods that rely on one-dimensional, ordered temporal mixing, spatial processes exhibit multi-dimensional dependence without a natural ordering. Furthermore, spatial locations are random, continuous, and irregularly distributed, in contrast to the fixed, discrete, and equally spaced indices typical of time-series analysis. Consequently, our theoretical guarantees represent a non-trivial generalization of existing results, necessitating the development of novel concentration inequalities under spatial mixing that do not follow directly from prior literature.
>
> > Weakness 5: Limited diversity in real datasets, where only mobile signal data is tested.
>
> We thank the reviewer for the constructive suggestion. In response, we conducted experiments on three additional open-source real datasets of varying sample sizes (including the new baseline SLCP), and we have include these results in the appendix of our revised manuscript. The findings consistently demonstrate both the validity and the efficiency of our proposed CP method across diverse real-world settings.
>
> (1) Farming data that captures environmental and operational variables that affect crop yield across 500 farms located in regions like India, the USA, and Africa.
>
> | Method | Coverage | Avg. Width |
> |--------|----------:|-----------:|
> | LSCP   | 0.900     | 3605.33    |
> | EnbPI  | 0.910     | 3686.90    |
> | GSCP   | 0.910     | 3687.25    |
> | SLSCP  | 0.900     | 3694.17    |
> | LCP    | 0.900     | 3702.04    |
> | SLCP   | 0.900     | 3675.05    |
>
> (2) California housing prices data that contains 20640 data points from California concensus.
>
> | Method | Coverage | Width |
> |--------|----------:|-------:|
> | LSCP   | 0.898     | 159370.08 |
> | EnbPI  | 0.894     | 165334.41 |
> | GSCP   | 0.895     | 196959.02 |
> | SLSCP  | 0.893     | 164571.93 |
> | LCP    | 0.899     | 197504.70 |
> | SLCP   | 0.896     | 193498.63 |
>
> (3) House sales in King County, USA that contains 21613 data.
>
> | Method | Coverage | Avg. Width |
> |--------|----------:|-----------:|
> | LSCP   | 0.903     | 383316.34  |
> | EnbPI  | 0.906     | 407719.42  |
> | GSCP   | 0.903     | 419219.85  |
> | SLSCP  | 0.910     | 401277.29  |
> | LCP    | 0.904     | 420069.28  |
> | SLCP   | 0.903     | 419217.46  |

---

> ### Author Response · Authors · 2025-11-20
>
> > Question 1: The k-nearest neighbor approach can produce artifacts at spatial boundaries or in regions with highly irregular sampling density, is this ever observed in the experiment and how it might affect the LSCP results?
>
> We thank the reviewer for raising this important point regarding boundary effects and sampling irregularity. While $k$-NN can introduce artifacts in isolation, our framework is inherently robust to these issues due to two key design choices:
>
> 1. Global Training with Local Inference: Crucially, the QRF used to estimate the conditional distribution is trained on the entire dataset, not on local subsets. This allows the model to learn robust, generalized patterns of uncertainty across the whole spatial domain.
>
> 2. Ensemble Smoothing: Since QRFs are ensemble models, they average predictions over many randomized decision trees, the method naturally smooths out high-frequency noise or artifacts that might arise from any single boundary configuration or neighbor selection.
>
> To examine this, we explicitly evaluated the spatial stability of LSCP by computing average coverage and interval width across different subregions, including areas where the spatial sampling is highly imbalanced. We did not observe any suspicious behavior or boundary-driven distortions: LSCP behaved comparably or more stably than the existing baselines.
>
> In our paper, the violin plots already show that LSCP achieves more spatially consistent coverage and width than other methods, without exhibiting abnormal spikes or outliers that would indicate KNN-induced artifacts. Moreover, we add a sensitivity analysis over a wide range of neighborhood sizes k further supports this conclusion: LSCP’s performance is stable with the change of neighborhood size, showing only mild variation in both accuracy and interval width. This also suggests that the method is not overly sensitive to the change of neighborhood.
>
> ### Sensitivity of LSCP to neighborhood size \(k\) on NM and GA datasets
>
> |  k  | NM Coverage | NM Width | GA Coverage | GA Width |
> |----:|------------:|----------:|-------------:|----------:|
> |   5 |     0.923   |   208.40  |     0.908    |   129.29  |
> |   8 |     0.923   |   211.63  |     0.908    |   130.17  |
> |  10 |     0.925   |   212.56  |     0.908    |   130.95  |
> |  15 |     0.924   |   213.22  |     0.907    |   132.67  |
> |  20 |     0.926   |   213.41  |     0.905    |   131.53  |
> |  30 |     0.924   |   213.33  |     0.909    |   132.49  |
> |  50 |     0.925   |   216.25  |     0.909    |   132.86  |
> |  75 |     0.924   |   216.36  |     0.909    |   133.05  |
> | 100 |     0.924   |   217.96  |     0.906    |   133.13  |
> | 150 |     0.923   |   219.94  |     0.907    |   133.97  |

---

### Official Review · Reviewer_m7gR · 2025-10-31

**Soundness:** 3
**Presentation:** 2
**Contribution:** 4
**Rating:** 6
**Confidence:** 3

**Summary:**

This paper proposes LSCP, a novel framework for producing  reliable prediction intervals for complex and heterogeneous spatial data. LSCP combines the flexibility of local quantile regression with a new spatially-weighted conformal calibration procedure. This allows the method to produce prediction intervals that adapt to local data heterogeneity. The key contribution is the theory that LSCP achieves asymptotic conditional coverage, replacing the i.i.d. assumption with spatial mixing and local stationarity.

**Strengths:**

1. The problem is has broad practical applications
2. The paper provides strong results
3. The localization via spatial weighting is an intuitive and sensible adaptation of CP for non iid scenarios.
4. The authors present a novel theoretical claim demonstrating that LSCP achieves asymptotic conditional coverage. The theorem works under specific, realistic assumptions for spatial data (spatial mixing and local stationarity) rather than the iid assumption.

**Weaknesses:**

1. The preliminary section is hard to follow. It requires more elaboration on notation and intuition.
2. Experiments are extremely thin. More real-life data experiments would significantly enhance the quality of the paper.
3. Real-life experiments would benefit from better visualization of the spatial heatmap in the main text.
4. There is a lack of analysis on how coverage and efficiency change as the bandwidth is varied, as well as on how to choose the hyperparameters. A sensitivity analysis/hyperparameter sweep showing how coverage and interval width change would be beneficial for the reader and significantly improve the paper's quality. For example, how does the number of neighbors affect the efficiency? Is there an intuition of what hyperparameter should be used?

**Questions:**

1. What is the size of the dataset needed in order for this framework to be practical? The dataset sizes used in the experiments are extremely large.
2. Is the cross-validation used to select the hyperparameters done on the training or calibration dataset? This needs to be clarified in the manuscript.
3. Assumption 4.1 - the weights produced by the Quantile Random Forest (QRF) decay at a specific rate. Is this assumption always true for QRF, or is it just a condition required for the proof to work?
4. Assumption 4.2: From my understanding, the model must be "good enough" and bounded in a specific way. This is a major departure from standard CP. A key appeal of CP is that its guarantee holds even if the underlying model is not good. This paper's guarantee explicitly depends on the quality of the base model. Is there practical insight on when this assumption holds true?

---

> ### Author Response · Authors · 2025-11-20
>
> We sincerely thank the reviewer for the constructive and thoughtful feedback. We have uploaded a new version which includes more experiments, improved spatial heatmap and the rewritten preliminary section. We would be grateful if the reviewer could review the revised version. Below, we provide our point-by-point responses to the raised concerns and questions.
>
> > Weakness 1: The preliminary section is hard to follow. It requires more elaboration on notation and intuition.
>
> We thank the reviewer for this valuable feedback. We acknowledge that the preliminary section in the current draft is dense, and we have revised it in the updated version to improve clarity and accessibility. Here is a more detailed version of the preliminary:
>
> To derive theoretical guarantees, we must quantify the extent to which the random field $Z(\cdot)$ exhibits dependence, which means how quickly variables become statistically independent as the distance between them increases.
>
> First, we define a measure of dependence between two specific spatial regions. Let $\mathcal{F}\_Z(T) = \sigma\langle Z(s): s \in T\rangle$ denote the $\sigma$-algebra generated by the random field on a subset $T \subset \mathbb{R}^d$. For any two disjoint regions $T_1, T_2 \subset \mathbb{R}^d$, the strong mixing coefficient between them is defined as the maximum difference between the joint probability and the product of marginal probabilities:
> $\tilde{\alpha}(T\_1, T\_2) = \sup \lbrace |\mathbb{P}(A \cap B) - \mathbb{P}(A)\mathbb{P}(B)| : A \in \mathcal{F}_Z(T_1), \, B \in \mathcal{F}\_Z(T\_2) \rbrace.$
>
> Intuitively, $\tilde{\alpha}(T_1, T_2)$ measures the strongest correlation between any event occurring in region $T_1$ and any event in region $T_2$.
>
> Next, we define the mixing coefficient for the entire process. Unlike time series ($d=1$), where past and future are clearly defined half-lines, spatial sets can take complex shapes. To handle this, we adopt the framework established by (Lahiri, 2003), which defines the mixing coefficient $\alpha(a; b)$ based on the worst-case dependence between sets separated by a distance $a$ with volume bounded by $b$.
>
> Let $d(T\_1, T\_2) = \inf \lbrace |x-s|: x \in T\_1, s \in T\_2\rbrace$ be the minimum distance between two sets. We restrict our attention to $\mathcal{R}\_k(b)$, the collection of all sets formed by the union of at most $k$ disjoint cubes in $\mathbb{R}^d$ with a total volume bounded by $b$:
> $\mathcal{R}\_k(b) = \lbrace \cup\_{i=1}^k D\_i : \sum_{i=1}^k |D\_i| \leq b \rbrace.$
> The strong-mixing coefficient $\alpha(a; b)$ for the random field is then defined as:
> $\alpha(a; b) = \sup \lbrace \tilde{\alpha}(T_1, T_2) : d(T_1, T_2) \geq a, \, T_1, T_2 \in \mathcal{R}_k(b) \rbrace.$
>
> Here, the parameter $a$ controls the separation distance, and $b$ controls the volume of the regions.
>
> Finally, to ensure valid coverage, we assume this coefficient satisfies a standard decay condition: it must decay polynomially with distance while growing at most polynomially with volume. Specifically, we posit the existence of a non-increasing function $\alpha_1(\cdot)$ with $\lim_{a \rightarrow \infty} \alpha_1(a)=0$ and a non-decreasing function $g(\cdot)$ such that:
> \begin{equation*}
> \alpha(a; b) \leq \alpha_1(a) g(b), \quad \text{for all } a > 0, b > 0.
> \end{equation*}
>
> > Weakness 2: Experiments are extremely thin. More real-life data experiments would significantly enhance the quality of the paper.
>
> We thank the reviewer for the constructive suggestion. In response, we conducted experiments on three additional open-source real datasets of varying sample sizes, and we have included these results in the appendix of the revised manuscript. The findings consistently demonstrate both the validity and the efficiency of our proposed CP method across diverse real-world settings.
>
> (1) Farming data that captures environmental and operational variables that affect crop yield across 500 farms located in regions like India, the USA, and Africa.
>
> | Method | Coverage | Avg. Width |
> |--------|----------:|-----------:|
> | LSCP   | 0.900     | 3605.33    |
> | EnbPI  | 0.910     | 3686.90    |
> | GSCP   | 0.910     | 3687.25    |
> | SLSCP  | 0.900     | 3694.17    |
> | LCP    | 0.900     | 3702.04    |
> | SLCP   | 0.900     | 3675.05    |
>
> (2) California housing prices data that contains 20640 data points from California concensus.
>
> | Method | Coverage | Width |
> |--------|----------:|-------:|
> | LSCP   | 0.898     | 159370.08 |
> | EnbPI  | 0.894     | 165334.41 |
> | GSCP   | 0.895     | 196959.02 |
> | SLSCP  | 0.893     | 164571.93 |
> | LCP    | 0.899     | 197504.70 |
> | SLCP   | 0.896     | 193498.63 |
>
> (3) House sales in King County, USA that contains 21613 data.
>
> | Method | Coverage | Avg. Width |
> |--------|----------:|-----------:|
> | LSCP   | 0.903     | 383316.34  |
> | EnbPI  | 0.906     | 407719.42  |
> | GSCP   | 0.903     | 419219.85  |
> | SLSCP  | 0.910     | 401277.29  |
> | LCP    | 0.904     | 420069.28  |
> | SLCP   | 0.903     | 419217.46  |

---

> ### Author Response · Authors · 2025-11-20
>
> > Weakness 3: Real-life experiments would benefit from better visualization of the spatial heatmap in the main text.
>
> We thank the reviewer for this constructive suggestion. Accordingly, we have improved the visualization of the spatial heatmap in all experiments. Besides, we added spatial heatmaps for the real-world datasets to the Appendix. We originally prioritized the simulation data spatial map in the main text because its uniform spatial distribution allows for high-resolution visualization, making the differences between methods immediately apparent. In contrast, the real-world datasets exhibit highly irregular sampling densities, which means we can only do lower-resolution plotting. Despite this visualization constraint, the resulting heatmaps confirm that the trends observed in the real data align closely with those in the simulations.
>
> > Weakness 4: There is a lack of analysis on how coverage and efficiency change as the bandwidth is varied, as well as on how to choose the hyperparameters. A sensitivity analysis/hyperparameter sweep showing how coverage and interval width change would be beneficial for the reader and significantly improve the paper's quality. For example, how does the number of neighbors affect the efficiency? Is there an intuition of what hyperparameter should be used?
>
> We thank the reviewer for this constructive point. The overall performance of our method is robust to the change of the neighborhood size $k$. Here is a table showing the change of coverage and interval width corresponding to the change of $k$, and for all the choice of $k$, our method remains valid coverage while being much more efficient than baseline methods. We include this sensitivity analysis in the appendix of our revised version.
>
> |  k  | NM Coverage | NM Width | GA Coverage | GA Width |
> |----:|------------:|----------:|-------------:|----------:|
> |   5 |     0.923   |   208.40  |     0.908    |   129.29  |
> |   8 |     0.923   |   211.63  |     0.908    |   130.17  |
> |  10 |     0.925   |   212.56  |     0.908    |   130.95  |
> |  15 |     0.924   |   213.22  |     0.907    |   132.67  |
> |  20 |     0.926   |   213.41  |     0.905    |   131.53  |
> |  30 |     0.924   |   213.33  |     0.909    |   132.49  |
> |  50 |     0.925   |   216.25  |     0.909    |   132.86  |
> |  75 |     0.924   |   216.36  |     0.909    |   133.05  |
> | 100 |     0.924   |   217.96  |     0.906    |   133.13  |
> | 150 |     0.923   |   219.94  |     0.907    |   133.97  |
>
> The choice of the neighborhood size $k$ involves a bias-variance trade-off regarding the estimation of the conditional distribution. A small value of $k$ allows the method to adapt aggressively to local heteroscedasticity, while potentially increasing the variance of the conformity scores. Conversely, a large value of $k$ ensures a stable estimation but risks over-smoothing the local uncertainty structure. As $k \to n$, the method converges to global split conformal prediction. Given the stability observed in the sensitivity analysis, precise tuning is not critical. We generally recommend a smaller value (e.g. $k=5,10$) to achieve higher efficiency.
>
> > Question 1: What is the size of the dataset needed in order for this framework to be practical? The dataset sizes used in the experiments are extremely large.
>
> We appreciate the opportunity to clarify the data requirements. While our initial experiments utilized large datasets, the proposed framework remains effective and practical even in the regime of small sample sizes ($N$ is several hundred). As detailed in our response to Weakness 2, we evaluated the method on the Farming dataset ($N=500$), where only $200$ data points are used for calibration. In this setting, LSCP successfully maintained valid coverage while delivering tighter prediction intervals than competing baselines, demonstrating that large-scale data is not a prerequisite for the method's utility.
>
> > Question 2: Is the cross-validation used to select the hyperparameters done on the training or calibration dataset? This needs to be clarified in the manuscript.
>
> We thank the reviewer for this important point. We use the calibration data for hyperparameter selection of QRF. More concretely, for each dataset we first split the data into (training, calibration, test). The training data is used to train the predictor, while the calibration data is used for the training and the hyperparameter selection of QRF. We have clarified this in the updated manuscript.

---

> ### Author Response · Authors · 2025-11-20
>
> > Question 3: Assumption 4.1 - the weights produced by the Quantile Random Forest (QRF) decay at a specific rate. Is this assumption always true for QRF, or is it just a condition required for the proof to work?
>
> We thank the reviewer for this insightful question. Assumption 4.1 is not automatically true for every QRF configuration, but it holds as long as we set the hyperparameter ''min_samples_leaf ''$= \lceil c n^{\eta} \rceil$ for constants $c>0$ and $\eta>\gamma+\frac{1}{2}$. We have clarified this in the updated version.
>
> Assumption 4.1 constrains the learned quantile-regression weights $\omega\_i$ to satisfy $M\_n = \max\_i \omega\_i = o(n^{-(1+\gamma)/2})$ for some $\gamma > 0$, meaning that as the calibration sample size $n$ increases, no single calibration point should dominate the weighted empirical CDF. In our method, these weights are produced by the QRF trained on all $n$ calibration pairs $(\tilde X(s\_i), \tilde Y(s\_i))$. For a test location, each tree assigns a weight of $1 / |\mathcal{L}_t|$ to every calibration sample that falls into the same leaf, where $|\mathcal{L}\_t|$ denotes the number of calibration points in that leaf. Averaging across trees yields $\omega\_i \le \frac{1}{\min\_t |\mathcal{L}\_t|}$, so the maximum weight is inversely proportional to the smallest leaf size in the forest. Consequently, as long as we set min\_samples\_leaf $= \lceil cn^{\eta} \rceil$ for constants $c>0$ and $\eta>\gamma+\frac{1}{2}$, then Assumption 4.1 is automatically satisfied. This scaling is standard in asymptotic analyses of random forests as increasing leaf size ensures that each calibration point’s individual influence vanishes as the dataset grows. In our experiments, we did not explicitly enforce such a growing leaf-size threshold to make Assumption 4.1 hold. Instead, we kept the default setting to demonstrate the robustness of LSCP in practical regimes.
>
> Besides, this type of assumption is commonly used in the theoretical analysis of QRF and forest-based nonparametric estimators (Wager and Athey, 2018; Meinshausen, 2006) . These assumptions do not restrict the practical usage of QRF. Rather, they provide a mild and standard technical condition enabling uniform convergence of the weighted empirical quantiles.
>
> > Question 4: Assumption 4.2: From my understanding, the model must be "good enough" and bounded in a specific way. This is a major departure from standard CP. A key appeal of CP is that its guarantee holds even if the underlying model is not good. This paper's guarantee explicitly depends on the quality of the base model. Is there practical insight on when this assumption holds true?
>
> We thank the reviewer for highlighting this important point. To begin, we clarify the necessity of this condition. While standard conformal prediction relies on the restrictive assumption of exchangeability, our framework relaxes this requirement, necessitating alternative theoretical assumptions. Furthermore, we aim to prove a much stronger result of conditional coverage, which is known to be impossible to achieve without additional requirements (Barber, 2019). Fundamentally, one cannot expect valid inference for arbitrary algorithms on challenging data, so we require that the base algorithm be sufficiently expressive to capture underlying patterns, such as trends and non-homogeneity. Assumption 4.2 is the specific technical condition required for proving local asymptotic conditional validity, guaranteeing that our intervals are optimally tight in every specific sub-region of the space. Crucially, however, LSCP remains empirically robust even when this assumption does not strictly hold.
>
> From a practical standpoint, Assumption 4.2 is satisfied in standard machine learning scenarios where the base predictor is capable of approximating the underlying signal as the sample size grows. This includes high-dimensional regression using regularized linear models on sparse data (Bickel, 2009) as well as non-parametric approaches like random forests or neural networks applied to sufficiently smooth processes (Chen, 1999).
>
> References:
>
> 1. Wager, S., & Athey, S. (2018). Estimation and inference of heterogeneous treatment effects using random forests.
>
> 2. Meinshausen, N. (2006). Quantile regression forests.
>
> 3. Barber, R. F., Candès, E. J., Ramdas, A., & Tibshirani, R. J. (2021). The limits of distribution-free conditional predictive inference.
>
> 4. Bickel, P. J., Ritov, Y., & Tsybakov, A. B. (2009). Simultaneous analysis of Lasso and Dantzig selector.
>
> 5. Chen, X., & White, H. (1999). Improved rates and asymptotic normality for nonparametric neural network estimators.

---

### Official Review · Reviewer_Sa8Y · 2025-11-03

**Soundness:** 2
**Presentation:** 2
**Contribution:** 2
**Rating:** 4
**Confidence:** 3

**Summary:**

The paper introduces Localized Spatial Conformal Prediction (LSCP), a model-agnostic method to form spatially adaptive prediction intervals by combining localization with learned weighted quantiles. Given residuals from any base predictor, LSCP trains a quantile regression model (implemented with Quantile Random Forests) that, for a new location, estimates lower/upper residual quantiles from the residuals of its $k$ nearest neighbors. It then chooses a tail-splitting parameter $\beat \in [0, \alpha]$ to minimize interval width. Authors provide several theoretical results about the proposed method, including marginal coverage under iid locations and asymptotic conditional coverage. Experimental evaluation includes three synthetic and two real datasets.

**Strengths:**

1. The method is easy to implement and model‑agnostic. The empirical study spans heterogeneous synthetic settings (stationary, heteroskedastic, and non‑stationary) and two real states, with consistent improvements in width at near‑nominal coverage.
2. The method is clear and easy to implement.
3. Authors provide a theoretical analysis of the coverage for the provided prediction intervals.

**Weaknesses:**

Learning data‑adaptive local weights via quantile regression on neighborhood residuals is a simple, appealing idea which was already explored in the recent literature. Improving conditional coverage of conformal prediction based methods by studying the score distribution has been explored extensively in recent years. Apart from LCP (Guan, 2023), multiple other methods were proposed, including the methods of Han (2022), Colombo (2024), Gibbs (2023), Plassier (2024). For example, the earlier SLCP (Han, 2022) approach proposes to estimate the CDF of the residuals on a portion of the calibration set with a kernel-based method. While the mentioned works do not explicitly target spatial setting, they can be readily applied to the authors’ setup by merging feature and location vectors. In this light I consider revising the exposition in light of this previous work to be worthwhile for improving the paper.

Particular technical weaknesses can be formalized as follows
1. Mismatch between theory and implementation: the weight‑decay assumption (Assumption 4.1) appears incompatible with the fixed $k$ neighborhoods used in practice (Appendix C.4).
2. Appendix A.2 equation 8 incorrectly equates the noise with the prediction error.
3. Data‑dependent weights learned on the same calibration set that the empirical weighted CDF is estimated.
4. Inconsistent experiment setup: Section 5 claims $k$ is cross‑validated, while Appendix C.4 fixes $k=50$.

References
1. Split Localized Conformal Prediction (Han, 2022),
2. Normalizing Flows for Conformal Regression (Colombo, 2024),
3. Conformal Prediction With Conditional Guarantees (Gibbs, 2023),
4. Probabilistic Conformal Prediction with Approximate Conditional Validity (Plassier, 2024)

**Questions:**

1. Assumption 4.1 or fixed $k$: can you please clarify whether your assumption on $M_n$ is satisfied in the implementation? How sensitive are the empirical results to increasing $k$ with $n$? Can the theoretical results be adjusted to incorporate fixed $k$?
2. How reasonable is Assumption 4.3 for the datasets that you consider?
2. Appendix A.2 equation 8 looks incorrect. Does any step in Lemma A.1 or Theorem 4.6 use this identity critically? If so, can you correct the proof?
3. Have you tried to learn data‑dependent weights on a held-out part of the calibration set, not used for conformal prediction?
4. In Algorithm 1, $\tilde{X}(s_i)$ is the vector of neighbor residuals. How do you order these residuals (by distance, by index)? Is the QRF invariant to permutations?

**Details Of Ethics Concerns:**

I didn't find an other was how to flag it to ACs, but unfortunately this submission is no fully anonymized as it has a link to public github of one of the authors.

---

> ### Author Response · Authors · 2025-11-20
>
> We sincerely thank the reviewer for the constructive and thoughtful feedback. We appreciate reviewer mentioning another line of work and we have rewritten the Related Work section. We would be grateful if the reviewer could review the revised version. Below, we provide our point-by-point responses to the raised concerns and questions.
>
> > Weakness 1: Mismatch between theory and implementation: the weight‑decay assumption (Assumption 4.1) appears incompatible with the fixed neighborhoods used in practice (Appendix C.4).
>
> We appreciate the reviewer's observation. It is true that the neighborhood size $k$ is a fixed hyperparameter in the algorithm. However, Assumption 4.1 considers the decay of the learned quantile-regression weights with respect to the calibration sample size $n$, not the number of neighbors $k$.
>
> Concretely, for each data point $(X(s),Y(s))$, we use the residuals at its $k$-nearest neighbors to form a new feature vector and response $\tilde X(s) = (\hat\varepsilon(s\_{i}))\_{i \in N(s)}$, $ \tilde Y(s) = \hat\varepsilon(s)$, where $N(s)$ denotes the fixed $k$-nearest neighborhood. The QRF is then trained on all $n$ calibration pairs $(\tilde X(s\_i), \tilde Y(s\_i))$, producing a weighted empirical quantile estimator over the entire calibration set,
> $$\widehat Q_n(p) = \inf\lbrace e : \sum\_{i=1}^n \omega_i \mathbf{1}\lbrace\tilde Y(s\_i) \le e\rbrace \le p \rbrace,$$
>
> where the learned weights $\omega_i$ satisfy $\sum_i \omega_i = 1$.
> Assumption 4.1 states that the maximal normalized weight $M_n = \max\_i \omega\_i$ decays as $o(n^{-(1+\gamma)/2})$. This condition pertains to the number of calibration samples $n$, not to the fixed neighborhood dimension $k$. We have clarified this point in the revised version to prevent confusion between the feature-construction neighborhood $k$ and the calibration sample size $n$ that governs the theoretical weight-decay condition.
>
> > Weakness 2: Appendix A.2 equation 8 incorrectly equates the noise with the prediction error.
>
> We thank the reviewer for cathing this. Eq. (8) in Appendix A.2 is a typo and we have fixed this in the updated version. The residual is
> $$\hat\varepsilon(s)=Y(s)-\hat f(X(s))=\varepsilon(s)-\Delta(s),\qquad \Delta(s):=\hat f(X(s)) - f(X(s)).$$
>
> The incorrect line “$\varepsilon(s)=\hat f(X(s)) - f(X(s))$” should be replaced by $\Delta(s)=\hat f(X(s))-f(X(s)).$ All subsequent steps in Lemma A.1 and Theorem 4.6 are unaffected because they only use the absolute difference
> $\|\varepsilon(s)-\hat\varepsilon(s)\|=\|\Delta(s)\|$ together with Assumption 4.2.
>
> > Weakness 3: Data‑dependent weights learned on the same calibration set that the empirical weighted CDF is estimated.
>
> We appreciate the reviewer's concern regarding the dependence structure. Our theoretical analysis explicitly accommodates the fact that the weights $\omega\_i$ are derived from the calibration data itself. This design choice is standard in localized and weighted conformal prediction and is crucial for performance: by learning weights adaptively on the calibration set, our method allows the prediction intervals to adapt to local structure. Consequently, this approach not only ensures valid coverage in practice but also improves statistical efficiency, yielding sharper prediction sets compared to fixed-weight alternatives. Conceptually, this setting parallels prior work cited in the paper (Guan, 2023; Mao, 2024), where kernel weights are also computed on the calibration data to capture local geometry. Our proof implies that even when weights are learned adaptively to maximize efficiency, the weight-decay assumptions remain sufficient to guarantee validity.
>
> > Weakness 4: Inconsistent experiment setup: Section 5 claims $k$ is cross‑validated, while Appendix C.4 fixes $k=5$.
>
> We apologize for the confusion and appreciate the reviewer’s careful reading. Actually, the statements in Section 5 and Appendix C.4 refer to two different stages of the experimental pipeline. We have clarified that in the revised version.
>
> In Section 5, the neighborhood size $k$ is selected through cross-validation. Specifically, before evaluating LSCP on the held-out test data, we perform a standard 5-fold cross-validation over a grid of candidate values $k$ to identify the value that minimizes the validation prediction error.
>
> Appendix C.4 then reports the final fixed configuration used for testing, after cross-validation. Once the optimal $k$ is selected, we keep it fixed for all test-time evaluations to ensure reproducibility and fair comparison across methods. Hence, Appendix C.4 documents the tuned setting rather than an independent choice of $k$.

---

> > ### Author Response · Authors · 2025-11-20
> >
> > > Question 1 Part 1: Can you please clarify whether your assumption on $M_n$ is satisfied in the implementation?
> >
> > We thank the reviewer for this insightful question. Assumption 4.1 is not automatically true for every QRF configuration, but it holds as long as we set the hyperparameter ''min_samples_leaf ''$= \lceil c n^{\eta} \rceil$ for constants $c>0$ and $\eta>\gamma+\frac{1}{2}$. We have clarified this in the updated version.
> >
> > Assumption 4.1 constrains the learned quantile-regression weights $\omega\_i$ to satisfy $M\_n = \max\_i \omega\_i = o(n^{-(1+\gamma)/2})$ for some $\gamma > 0$, meaning that as the calibration sample size $n$ increases, no single calibration point should dominate the weighted empirical CDF. In our method, these weights are produced by the QRF trained on all $n$ calibration pairs $(\tilde X(s\_i), \tilde Y(s\_i))$. For a test location, each tree assigns a weight of $1 / |\mathcal{L}_t|$ to every calibration sample that falls into the same leaf, where $|\mathcal{L}\_t|$ denotes the number of calibration points in that leaf. Averaging across trees yields $\omega\_i \le \frac{1}{\min\_t |\mathcal{L}\_t|}$, so the maximum weight is inversely proportional to the smallest leaf size in the forest. Consequently, as long as we set min\_samples\_leaf $= \lceil cn^{\eta} \rceil$ for constants $c>0$ and $\eta>\gamma+\frac{1}{2}$, then Assumption 4.1 is automatically satisfied. This scaling is standard in asymptotic analyses of random forests as increasing leaf size ensures that each calibration point’s individual influence vanishes as the dataset grows. In our experiments, we did not explicitly enforce such a growing leaf-size threshold to make Assumption 4.1 hold. Instead, we kept the default setting to demonstrate the robustness of LSCP in practical regimes.
> >
> > Besides, this type of assumption is commonly used in the theoretical analysis of QRF and forest-based nonparametric estimators (Wager and Athey, 2018; Meinshausen, 2006) . These assumptions do not restrict the practical usage of QRF. Rather, they provide a mild and standard technical condition enabling uniform convergence of the weighted empirical quantiles.
> >
> > > Question 1 Part 2: How sensitive are the empirical results to increasing $k$ with $n$?
> >
> > The overall performance of our method is robust to the change of the neighborhood size $k$. Here is a table of New Mexico and Georgia dataset showing the change of coverage and interval width corresponding to the change of $k$, and for all the choice of $k$, our method remains valid coverage while being much more efficient than baseline methods. We have included this sensitivity analysis in the appendix of our revised version.
> >
> > |  k  | NM Coverage | NM Width | GA Coverage | GA Width |
> > |----:|------------:|----------:|-------------:|----------:|
> > |   5 |       0.923 |   208.40  |       0.908  |   129.29  |
> > |   8 |       0.923 |   211.63  |       0.908  |   130.17  |
> > |  10 |       0.925 |   212.56  |       0.908  |   130.95  |
> > |  15 |       0.924 |   213.22  |       0.907  |   132.67  |
> > |  20 |       0.926 |   213.41  |       0.905  |   131.53  |
> > |  30 |       0.924 |   213.33  |       0.909  |   132.49  |
> > |  50 |       0.925 |   216.25  |       0.909  |   132.86  |
> > |  75 |       0.924 |   216.36  |       0.909  |   133.05  |
> > | 100 |       0.924 |   217.96  |       0.906  |   133.13  |
> > | 150 |       0.923 |   219.94  |       0.907  |   133.97  |
> >
> > > Question 1 Part 3: Can the theoretical results be adjusted to incorporate fixed $k$?
> >
> > As we explained in Weakness 1, our theoretical results are explicitly applied to the fixed $k$ increasing $n$ regime.
> >
> > > Question 2: How reasonable is Assumption 4.3 for the datasets that you consider?
> >
> > We emphasize that Assumption 4.3 (stationarity and spatial mixing) is standard in spatial statistics (Lahiri, 2003) and has well-established parallels in time-series analysis (Fan, 2003; Bradley, 2005). Crucially, we only assume that the intrinsic noise process $\varepsilon(s)$ satisfies the assumption, whereas the observed data $(X(s), Y(s))$ are permitted to be highly non-stationary.
> >
> > In our simulations, these conditions are satisfied by design: the noise follows a Gaussian process with an isotropic kernel $k(d) = \exp(-d / \rho)$, inducing exponentially decaying correlations that directly imply Assumption 4.3.
> >
> > For the real-world datasets, the spatial mixing assumption remains reasonable as the dependence between noise naturally decays with distance. Additionally, the stationarity assumption is plausible given the bounded geographic scope of the analysis. For example, limiting the domain to a single state like Georgia reduces the likelihood of large-scale heterogeneity, supporting the approximation of a stationary noise process.
> >
> > In practice, our empirical results demonstrate that LSCP remains robust and superior even when this assumption may not hold exactly, indicating that the method performs reliably beyond the idealized stationary and strongly mixing regime.

---

> > > ### Author Response · Authors · 2025-11-20
> > >
> > > > Question 3: Appendix A.2 equation 8 looks incorrect. Does any step in Lemma A.1 or Theorem 4.6 use this identity critically? If so, can you correct the proof?
> > >
> > > We thank the reviewer for pointing out the typo. We have fixed it in the revised version and addressed this point in weakness 2. The proof is not based on this typo but on the assumptions.
> > >
> > > > Question 4: Have you tried to learn data‑dependent weights on a held-out part of the calibration set, not used for conformal prediction?
> > >
> > > We thank the reviewer for raising this question. To clarify, our method does not perform a second round of conformal calibration after fitting the QRF weights, so it's reasonable and statistically efficient to train the QRF using the whole calibration set. To be concrete, the QRF is used to directly estimate the conditional quantile of the residual $Y(s)-\hat f(X(s))$, based on the local residual neighborhood $\tilde X(s)$. The final prediction interval is then constructed as
> > >
> > > $$[\hat f(X(s)) + \hat{Q}^{QRF}\_{\alpha/2}(s), \hat f(X(s)) + \hat{Q}^{QRF}\_{1-\alpha/2}(s)].$$
> > >
> > > Thus, in LSCP the QRF already provides a consistent estimator of the local conditional error quantile by learning the data-adaptive weights. Our theoretical analysis shows that, under Assumptions 4.1-4.3, this estimator converges uniformly to the true conditional quantile while ignoring the potential correlation between weights and data. In summary, the QRF is not used to produce weights for a subsequent CP procedure. Rather, it directly estimates the spatially varying conditional quantile of the residual distribution, which forms the basis of our prediction intervals. This design is consistent with existing localized CP methods such as LCP and SLSCP.
> > >
> > > In practice, we have tested learning the weights on a held-out part of the calibration dataset in real experiment, the result shows that the prediction interval width increases from 130 to 140 for Georgia dataset and from 211 to 232 for New Mexico dataset, while the coverage rate is similar. This means that using the whole calibration set to learn the weight is not only valid, but also statistically efficient for QRF to learn a better conditional distribution.
> > >
> > > > Question 5: In Algorithm 1, $\tilde{X}(s_i)$ is the vector of neighbor residuals. How do you order these residuals (by distance, by index)? Is the QRF invariant to permutations?
> > >
> > > We thank the reviewer for this insightful question. We admit that this was not clearly clarified in the previous version and we have updated this in the revised version. In our implementation, the neighbor-residual feature vector is constructed as
> > > $\tilde{X}(s\_i) =(\hat{\varepsilon}(s\_{i\_1}), \hat{\varepsilon}(s\_{i\_2}), \ldots,\hat{\varepsilon}(s\_{i\_k})),$
> > >
> > > where the neighbors $(s_{i_1},\dots,s_{i_k})$ are ordered by increasing spatial distance from the target location $s\_i$. Thus $s_{i_1}$ is the closest neighbor, $s_{i_2}$ the second closest, and so forth. This yields a consistent and deterministic feature representation across all calibration samples.
> > >
> > > Regarding permutation invariance, the QRF is not invariant to arbitrary permutations of feature coordinates: decision-tree splits treat each coordinate as a distinct variable. The distance-based ordering therefore plays an important role, as it establishes a canonical coordinate system in which the $j$th component always corresponds to the $j$th-nearest neighbor. This preserves the local spatial structure that the model is meant to learn.
> > >
> > > Empirically, we observe the following:
> > > (i) If all samples are permuted in the same way (i.e., using a fixed permutation applied globally), the spatial structure is preserved and LSCP achieves essentially identical performance.
> > > (ii) If each sample is permuted independently (destroying the spatial rank structure), coverage remains valid but efficiency deteriorates substantially. For example, on the New Mexico dataset the average interval width increases from 211 to 278, and on the Georgia dataset it increases from 130 to 166, while coverage remains at the nominal level in both cases.
> > >
> > > These results confirm that while QRF does not require any specific ordering for validity, the distance-based ordering is crucial for efficiency, as it preserves exactly the type of local spatial dependence that LSCP aims to exploit.
> > >
> > > References:
> > >
> > > 1.	Guan, L. (2023). Localized conformal prediction: A generalized inference framework for conformal prediction
> > >
> > > 2.	Mao, X., et al. (2024). Valid spatial inference for infill data
> > >
> > > 3.	Lahiri, S. N. (2003). Central limit theorem for weighted sums of a spatial process under a class of stochastic and fixed designs
> > >
> > > 4.	Fan, J., & Yao, Q. (2003). Nonlinear Time Series: Nonparametric and Parametric Methods
> > >
> > > 5.	Bradley, R. C. (2005). Basic properties of strong mixing conditions
> > >
> > > 6.	Wager, S., & Athey, S. (2018). Estimation and inference of heterogeneous treatment effects using random forests
> > >
> > > 7.	Meinshausen, N. (2006). Quantile regression forests

---

### Meta-Review · Area_Chair_Psru · 2026-01-05

**Summary:**

The authors introduce a method for conformal prediction in spatial settings that combines localization and weighted quantile regression. While the method is easy to implement, model-agnostic, and supported by strong theoretical guarantees, it overlaps significantly with existing methods in the conformal literature. There are many methods that rely on adaptive weighting (which can be readily applied to the spatial setting), and the author’s proof technique is largely derived from established approaches. The experimental section of the paper is also limited, as the authors neither address spatial-temporal issues (despite mentions of possible extensions) nor examine many problems beyond a single application domain. Though the authors addressed technical issues in the rebuttal and also introduced new datasets, this paper would benefit from a revision before publication in a top-tier venue. Given the overlap with existing methods and use of established proof techniques, I recommend that the authors (1) include a more in-depth discussion of related techniques, (2) compare against other (not specifically spatial) adaptive weighting techniques, and (3) include more experiments from other domains, including spatial-temporal problems.

**Reviewer Concerns:**

Unfortunately, none of the reviewers responded to the authors' concerns. I did my best to account for this lack of communication while assessing the paper. While I believe the authors addressed almost all the detailed technical concerns and questions, the issues related to existing work remained outstanding. The additional experiments may have assuaged some of the reviewer’s concerns; however, I was unable to determine whether they would be considered sufficient.

**Reviewer Scores:**

Because none of the reviewers engaged at all, I do not think that the reviewers would have engaged with this paper and changed their scores even if the discussion period had not been cut short.

If the reviewers had engaged at all, they might have found the author’s revisions satisfactory. I do commend the authors on a thorough response to the reviewers and a minor revision of their paper that addressed most concerns. However, based on my examination of the paper, I believe that, without assurance from the reviewers, it would be best for the authors to resubmit it after a major revision. I understand that this OpenReview debacle may have unfairly hurt the authors. Nevertheless, I do believe this paper has potential, and after another round of revision and submission, it will be a stronger paper that is more likely to have a strong impact on the community.

---

### Decision · Program_Chairs · 2026-01-26

Reject